# Electricity generation from carbon dioxide adsorption by spatially nanoconfined ion separation

Zhuyuan Wang [1,2], Ting Hu[2], Mike Tebyetekerwa [1], Xiangkang Zeng [1], Fan Du[2], Yuan Kang[2], Xuefeng Li[1], Hao Zhang[1], Huanting Wang [2] & Xiwang Zhang [1,2,3] ✉

Selective ion transport underpins fundamental biological processes for efficient energy conversion and signal propagation. Mimicking these 'ionics' in synthetic nanofluidic channels has been increasingly promising for realizing self-sustained systems by harvesting clean energy from diverse environments, such as light, moisture, salinity gradient, etc. Here, we report a spatially nanoconfined ion separation strategy that enables harvesting electricity from $CO_2$ adsorption. This breakthrough relies on the development of Nanosheet-Agarose Hydrogel (NAH) composite-based generators, wherein the oppositely charged ions are released in water-filled hydrogel channels upon adsorbing $CO_2$. By tuning the ion size and ion-channel interactions, the released cations at the hundred-nanometer scale are spatially confined within the hydrogel network, while ångström-scale anions pass through unhindered. This leads to near-perfect anion/cation separation across the generator with a selectivity $(D^-/D^+)$ of up to $1.8 \times 10^6$, allowing conversion into external electricity. With amplification by connecting multiple as-designed generators, the ion separation-induced electricity reaching 5 V is used to power electronic devices. This study introduces an effective spatial nanoconfinement strategy for widely demanded high-precision ion separation, encouraging a carbon-negative technique with simultaneous $CO_2$ adsorption and energy generation.

Global warming caused by greenhouse gas emissions[1–4], primarily carbon dioxide ($CO_2$) emissions from human activities[4–6], is one of the key challenges of our time. Carbon capture, utilization, and storage (CCUS) strategies are expected to play crucial roles in mitigating this challenge. However, current CCUS technologies remain energy-intensive and costly, compromising their environmental and economic sustainability[7–9]. At present, chemical-based $CO_2$ adsorption, where $CO_2$ is adsorbed by aquatic adsorbents like alkanolamines or amines, is the most commercially available option[10]. The negative enthalpy change ($\Delta H$) of the chemical adsorption process, as indicated

by its spontaneous and exothermic nature, offers the potential for serving as an energy harvesting target to enhance its energy efficiency[5,8].

Choosing a feasible approach for realizing the energy harvesting from $CO_2$ can be guided by nature-based 'ionics' processes[11]. Biological processes of energy conversion or signal propagation are fundamentally reliant on regulated ion transport in bio-channels[11–13]. This has encouraged emerging advances in pursuing selective ion transport in synthetic nanofluidic channels for harvesting electricity from unconventional resources[14]. Devices of this technique involve creating

[1]UQ Dow Centre for Sustainable Engineering Innovation, School of Chemical Engineering, The University of Queensland, Queensland, St Lucia, Australia. [2]Department of Chemical and Biological Engineering, Monash University, Clayton, Australia. [3]ARC Centre of Excellence for Green Electrochemical Transformation of Carbon Dioxide (GETCO2), Brisbane, Australia. ✉e-mail: xiwang.zhang@uq.edu.au

directional ion separation across the artificial channels in response to external environmental gradients exemplified by thermal[15], pressure[16,17], salinity[18,19], moisture[20–22], light[23], etc. The external environment drives charged ions to diffuse across the ion channels presented in the generators. The ion channels are pre-designed to transport oppositely charged ions at distinguishable speeds, leading to the generation of net diffusion current and electric potential difference across the generator[21,24].

In commercially available $CO_2$ chemical adsorption, the amine solutions can generate oppositely charged ions upon $CO_2$ absorption as follows: $CO_2 + R\text{-}NH_2 + H_2O \rightleftharpoons R\text{-}NH_3^+ + HCO_3^-$ (Equation 1), if these ions are effectively separated, they also suitable for electricity generation as demonstrated recently[8,25,26]. However, the positively charged amine ions and negatively charged bicarbonate ions generated in the conventional $CO_2$ adsorption process possess similar sizes at the molecular level and are well mixed, making their efficient separation challenging[10,27,28]. Dependence solely on a positively charged separation membrane yielded a mobility selectivity of 2.5 between $HCO_3^-$ and $R\text{-}NH_3^+$ ions[26], a significantly lower magnitude compared to what was attained in the osmotic energy generation process[19]. Addressing this challenge could potentially facilitate the pursuit of efficient energy conversion and sufficient power density in this promising field.

Here, we report a NAH composite-based generator that can realize precise ion separation across its internal channels for generating electricity directly from adsorbing $CO_2$ (Fig. 1a). To overcome the size similarity of the generated ions by traditional adsorbents, nanocomposite adsorbents are synthesized by grafting polyethyleneimine (PEI) molecules with abundant $\text{-}NH/\text{-}NH_2$ groups for $CO_2$ adsorption onto two-dimensional (2D) hexagonal boron nitride (h-BN) nanosheets, forming PEI-functionalized h-BN nanosheets (h-BN-$NH_2$) (Fig. 1b i, Supplementary Fig. 1). Upon adsorbing $CO_2$, the h-BN-$NH_2$ nanocomposites release positively charged $NH_3^+$ ions bonded on h-BN possessing sizes at hundred nanometer scale (100–300 nm) and negative Å-scale free $HCO_3^-$ ions in the hydrogel channels, creating a size disparity between them to over two magnitudes (Fig. 1b ii). Since the extended h-BN skeleton of the positive ions is trapped by the agarose hydrogel matrix while the negative $HCO_3^-$ ions are left freely in the interconnected water-filled channels, these oppositely charged ions diffuse across the generator at a remarkable rate difference ($>10^6$). (Fig. 1b iii). Such a diffusion rate gap is translated into electricity and further amplified to power external electronic devices. This study utilizes the gaseous $CO_2$ as an energy-harvesting target. The demonstrated ion engineering strategies could advance other ion-separation-related energy, resource, and environmental applications.

## Results and discussion
### Rational design and fabrication of the electricity generator
h-BN-$NH_2$ nanosheets, as a crucial component of the NAH electricity generator, were obtained through a sticky exfoliation method as reported in our previous work[29]. The process involved ball milling of layered h-BN and ultrahigh-viscosity liquid PEI to achieve simultaneous exfoliation and functionalization. The high viscosity of PEI efficiently breaks and delaminates layered bulk h-BN particles, and meanwhile PEI molecules with reactive-ready amine groups are grafted onto the freshly exfoliated h-BN nanosheets through mechanochemical reactions[30]. The resulting product is named as h-BN-$NH_2$, which was collected and rinsed with water to remove the free PEI molecules. The thick h-BN flakes that were not sufficiently exfoliated were removed through the process of centrifugation. Transmission electron microscopy (TEM) images show that the resultant h-BN nanosheets have a lateral size in the range of from ~100 to 300 nm (Fig. 1c, supplementary Figs. 2, 3). The thermogravimetric and elemental analysis also reveal that the PEI molecules bonded to h-BN nanosheets account for 7.92 wt% of the final product (Supplementary Table 1, Supplementary Fig. 4).

The crystalline structure of BN lattices was preserved even after exfoliation and PEI grafting via chemical bonding (Supplementary Fig. 5), as shown by the characteristic X-ray diffraction peaks at 26.2° and 42.8°, corresponding to (002) and (100) planes of hexagonal boron nitride (Supplementary Fig. 6). Moreover, the X-ray diffraction (XRD) peaks of exfoliated h-BN show a significantly reduced intensity and broadened width, indicating successful exfoliation in terms of lateral size and layer number of h-BN.

To create artificial ion channels for electricity generator, agarose hydrogel, which contains ~98 wt% of water, was selected as the matrix due to its highly porous structure with suitable pore size which is large enough to transport bicarbonate ions but too small for h-BN-$NH_3^+$ nanosheets to penetrate through freely. The mixture solution of h-BN-$NH_2$ nanosheets and agarose was drop-casted into a polydimethylsiloxane (PDMS) mold and underwent a thermo-responsive sol-gel transition to produce a flexible hydrogel film (Fig. 1d, e). The ionic PEI molecules grafted on h-BN nanosheets improve the dispersibility of nanosheets in agarose solution in comparison to neat hydrophobic h-BN[30] (Fig. 1d, e, Supplementary Fig. 7). The agarose molecules form a three-dimensional interconnected reticular matrix with an average diameter of 102 nm, resulting from the extended hydrophilic polymer chains (Fig. 1d, Supplementary Fig. 3). Inter/intra-chain crosslinking was established by weaving the functional groups of (-OH) of agarose molecules into a coordinated hydrogen bonding network[31]. The unique properties of the hydrogel matrix provide strong mechanical strength and high flexibility in shape. As shown in Fig. 1a, to construct the prototype NAH electricity generator, two NAH hydrogels, serving as ion releasing and receiving reservoirs, respectively, are connected with a pure agarose hydrogel which acts as high-speed ion-selective channels.

### Electricity generation from $CO_2$ adsorption
To evaluate the electricity generation capacity of NAH electricity generator, it was placed into a testing box (50 by 25 cm, Supplementary Fig. 8) with a $CO_2$ inlet. The ion receiving reservoir side was covered while the ion-releasing reservoir side was exposed to $CO_2$ gas. To start the testing, $CO_2$ was fed into the testing box at a rate of 2.5 L min$^{-1}$ for 3 min. Upon $CO_2$ feeding, the open-circuit voltage ($V_{OC}$) of the NAH electricity generator peaked at approximately $80 \pm 10$ mV within 1 to 2 h (Fig. 2a), and then gradually decreased to an undetectable level in about 15 h, confirming that electricity was generated upon $CO_2$ adsorption. Gas $CO_2$ was then fed into the box again to start another test cycle. The electricity generation was found to be robust, with the peak $V_{OC}$ maintained at a level of $80 \pm 10$ mV in five consecutive tests spanning 60 h. The short-circuit current ($I_{SC}$) behaved similarly to $V_{OC}$, peaking at approximately $150 \pm 20$ nA for 1 to 2 h before it slowly faded to 0 in the following 5 to 10 h (Fig. 2b). However, unlike $V_{OC}$, the peak $I_{SC}$ decayed after each $CO_2$ adsorption cycle, and the peak $I_{SC}$ of the 5$^{th}$ cycle was only half that of the 1$^{st}$ cycle. For $V_{OC}$ testing, the total current was set as 0, which equals to an infinite external load resistance (R to ∞) while the total resistance of the $I_{SC}$ testing is only contributed by the intrinsic resistance of the generator. Thereby, the lower total resistance during $I_{SC}$ testing led to higher discharging current and thus an apparent performance decay relative to that of $V_{OC}$ testing[20,24].

The observed $I_{SC}$ decay necessitates a proper regeneration method to ensure the reusability of NAH electricity generator. Referring to the Bjerrum plot, the adsorbed $CO_2$ presents mainly as $HCO_3^-$ ions in the pH range from 6.0 to 9.0. Considering the aqueous environment of the NAH electricity generator, the commonly adopted pH-swing strategy should be able to remove $CO_2$ and regenerate NAH electricity generator[5,8]. During the pH swing desorption process, $HCO_3^-$ ions can transition into $CO_3^{2-}$ ions if the pH value is increased above 10.0, allowing sequestration by reacting with $Ca^{2+}$ to form $CaCO_3$ precipitates[25]. Alternatively, lowering the pH value below 4.0 converts

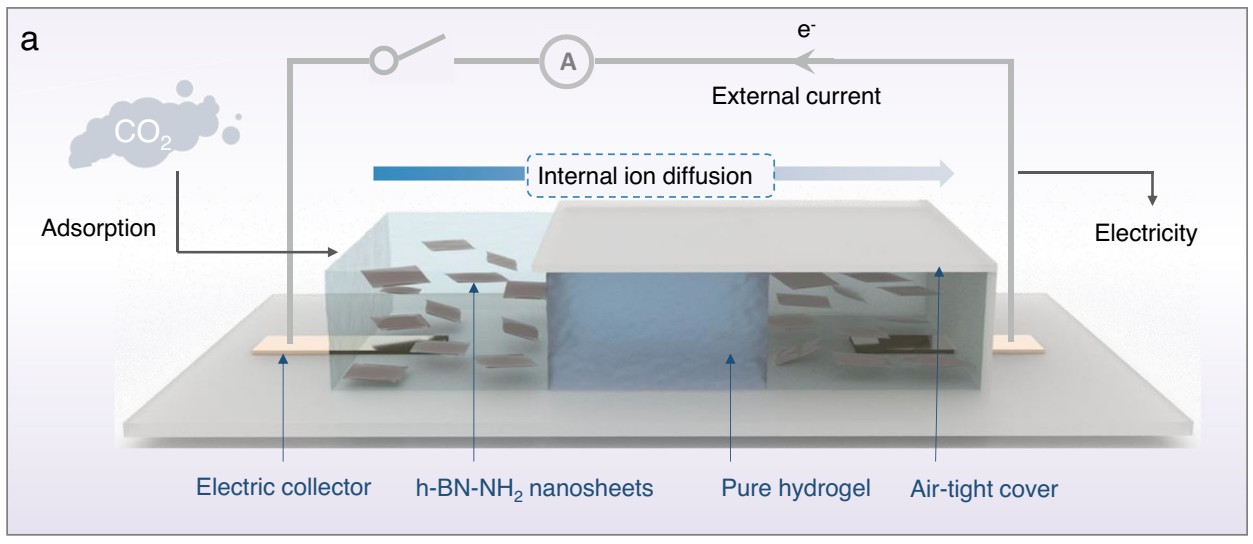

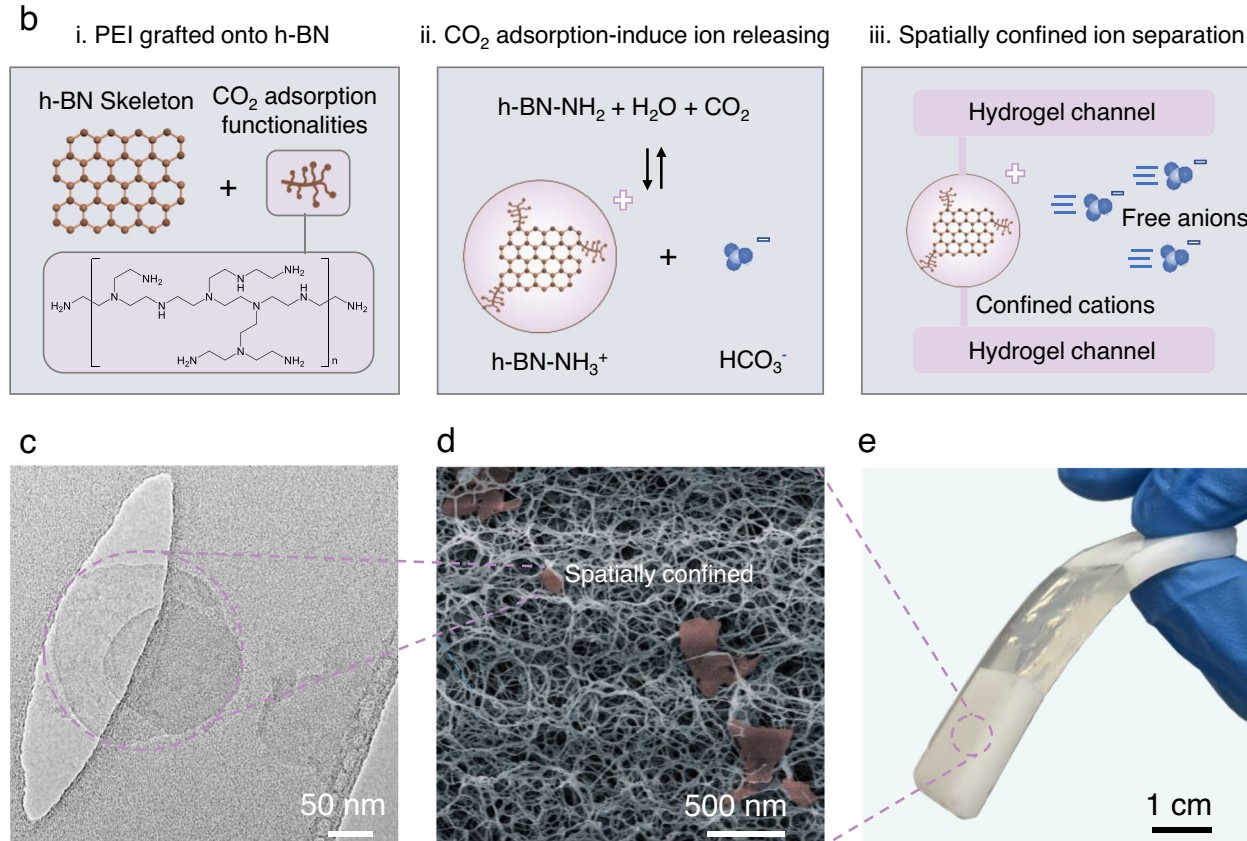

**Fig. 1 | Rational design of the NAH electricity generator from $CO_2$ adsorption.** **a** Schematic illustration of the prototype NAH electricity generator. The generator was intentionally designed with a symmetric structure between two electrical collectors to ensure a balanced chemical potential across the generator at the initial stage. **b** The process of ion separation induced electricity generation from $CO_2$ adsorption, it starts from (i) grafting PEI onto boron nitride nanosheets to get h-BN-NH$_2$ adsorbents, (ii) the $CO_2$ adsorption induces the ions releasing of cations (h-BN-NH$_3^+$) and anions (HCO$_3^-$), (iii) the cations in hundred nanometer are spatially confined while the anions can freely transverse within the hydrogel channels, leading to precise ion separation. **c** TEM image of the as-exfoliated and functionalized h-BN-NH$_2$ nanosheets. **d** SEM image of the cross-section structure of the NAH composite. The red discs are the trapped h-BN-NH$_2$ nanosheets in hydrogel network. **e** Photograph of the NAH electricity generator.

HCO$_3^-$ ions into $CO_2$ gas, which can be harvested for utilization[32]. The regeneration was conducted by immersing the generator in an alkaline buffer solution (pH=10). The outcomes demonstrate that the $I_{SC}$ and $V_{OC}$ signals can be fully restored to their initial levels via pH swing after five adsorption-and-discharging cycles (refer to the last regeneration cycle in Fig. 2b and Supplementary Fig. 9).

Electricity generation of NAH electricity generator can be easily boosted by increasing the density of $CO_2$ adsorption sites in NAH generator through increasing nanosheet concentration or raising PEI grafting ratios of h-BN-NH$_2$. Increasing the concentration of h-BN-NH$_2$ nanosheets in NAH hydrogel from 1 mg mL$^{-1}$ to 5 mg mL$^{-1}$ resulted in an increase of $V_{OC}$ from 90.4 mV to 145.7 mV with an extended half-value

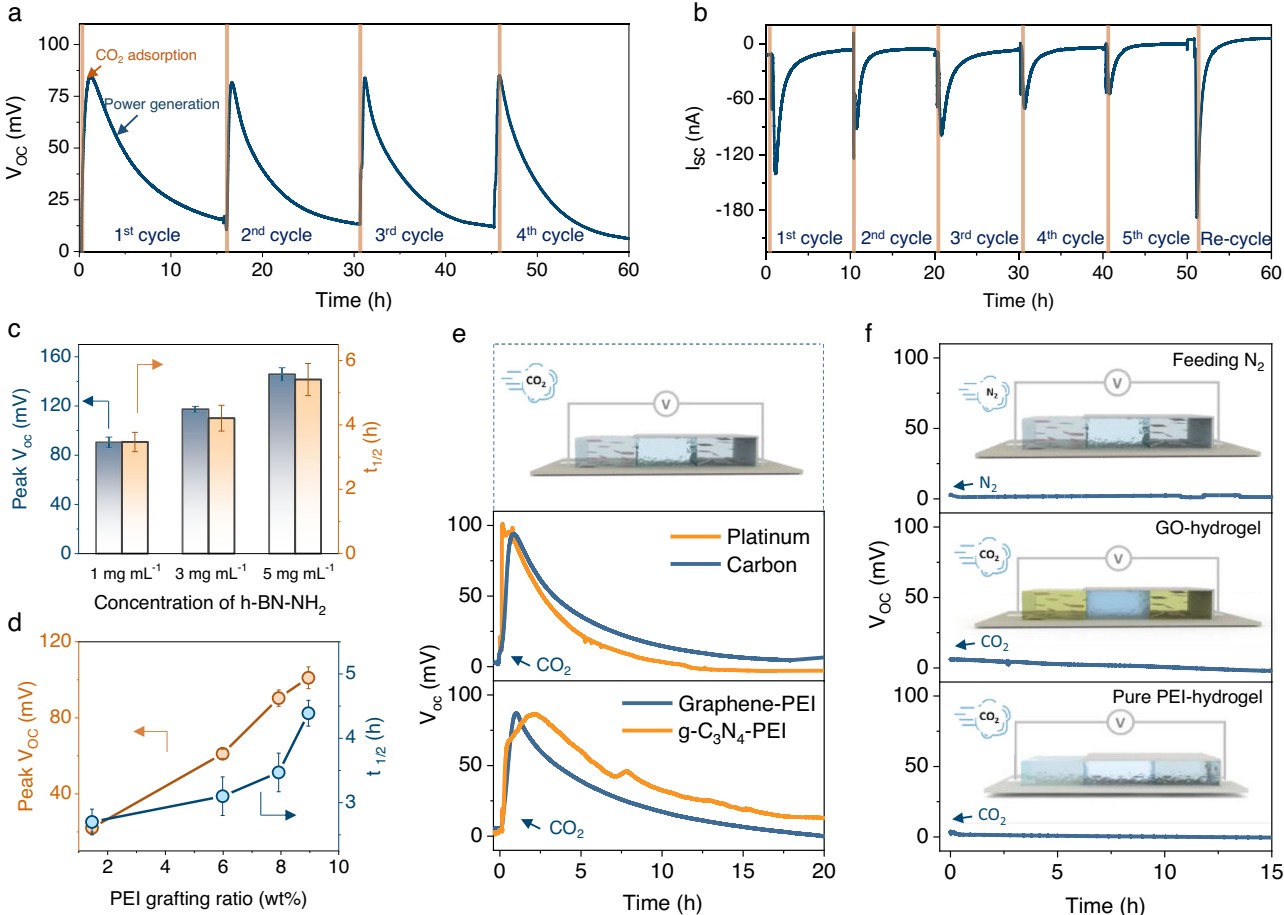

**Fig. 2 | CO₂-induced electricity generation of NAH electricity generator and the effects of key factors on performance. a** Cyclic $CO_2$ adsorption-induced open circuit voltage of NAH electricity generator. **b** Cyclic $CO_2$ adsorption-induced short circuit current of NAH electricity generator. A pH-swing was applied to recover the power generation after the 5th cycle. The $CO_2$ was fed into the testing box at a rate of 2.5 L min⁻¹ for 3 min in each cycle for both $V_{OC}$ and $I_{SC}$ tests. **c** Electricity generation in terms of the peak $V_{OC}$ and half value time ($t_{1/2}$) of the induced open circuit potential as a function of the concentration of the h-BN-NH₂ nanosheets in agarose hydrogel. **d** Electricity generation in terms of the peak $V_{OC}$ and half value time ($t_{1/2}$) of the induced open circuit potential as a function of the weight ratio of PEI functionalities on h-BN nanosheets. Error bars are standard deviations from three tests. **e** Electricity generation using carbon and platinum electrodes for electrical signals collection (middle panel), and Electricity generation using PEI functionalized graphene (Graphene-PEI) and PEI functionalized graphic carbon nitride (g-C₃N₄-PEI) (bottom panel). **f** Electricity generation in different control experiments: using N₂ to substitute CO₂ as the feed gas for the testing (top panel), replacing the h-BN nanosheets in the generator with graphene oxide nanosheet (middle panel) and pure PEI molecules (bottom panel).

period of $V_{oc}$ from 3.5 h to 5.4 h (Fig. 2c). No further significant enhancement was achieved by increasing the concentration to 10 mg mL⁻¹ (Supplementary Fig. 10). Raising the PEI grafting ratios of h-BN-NH₂ nanosheets from 1.46 wt.% to 8.95 wt.% boosted $V_{oc}$ by five times, and the half-value period was also extended by over two times (Fig. 2d).

To gain a better understanding of the origin of electricity generation, NAH electricity generator was tested under various conditions. First, the silver electrodes of the electricity generator were replaced by inert carbon and platinized titanium electrodes, respectively. The devices using inert electrodes still generated electricity with a peak $V_{oc}$ of ~90 mV (Fig. 2e). These rule out the possibility that electricity comes from the chemical reactions on the electrodes. Second, when we replaced CO₂ with N₂ as the feed gas, no obvious electricity signals were detected (Fig. 2f), indicating that electricity was generated from CO₂ adsorption.

To investigate the importance of the 2D skeletons of h-BN-NH₂ nanosheets in the hydrogel matrix, we tailored the components of NAH hydrogel. When PEI-functionalized graphene and graphitic carbon nitride (g-C₃N₄), which have similar structures to h-BN-NH₂ nanosheets, were applied as nanofillers in hydrogel, the as-constructed generators had similar performance to the h-BN-NH₂ generator

(Fig. 2e). However, when graphene oxide (GO) nanosheets with -OH/-COOH groups were used, no electricity was generated under identical operation conditions, highlighting the critical role of -NH/-NH₂ groups (Fig. 2f). In addition, when pure liquid PEI was used to replace 2D h-BN-NH₂ nanosheets, the generator failed to generate electricity either (Fig. 2f). These results reveal that large physical size of nanosheets (hundreds of nanometers) and abundant amino groups are essential for the electricity generation in NAH electricity generator.

The CO₂ adsorption capacity of h-BN-NH₂ nanosheets was measured as shown in Fig. 3a. The results reveal that h-BN-NH₂ nanosheets grafted with 7.92 wt.% of PEI functionalities exhibit a CO₂ adsorption capacity of 0.238 mmol g⁻¹ at standard temperature and pressure, while bulk h-BN exhibits no measurable CO₂ adsorption capacity. To confirm the production of equimolar amine and bicarbonate ions, the pH change of h-BN-NH₂ water solution during CO₂ adsorption process was monitored, showing a rapid decrease from 8 to 4.5 in minutes (Fig. 3b). The release of ions from adsorption is further substantiated by a threefold higher ion conductivity observed in the nanosheet solution bubbled with CO₂ compared to N₂. (Fig. 3c and Supplementary Note 3).

Selective directional ion transport is required following the release of ions to achieve diffusion current. The pore size of the

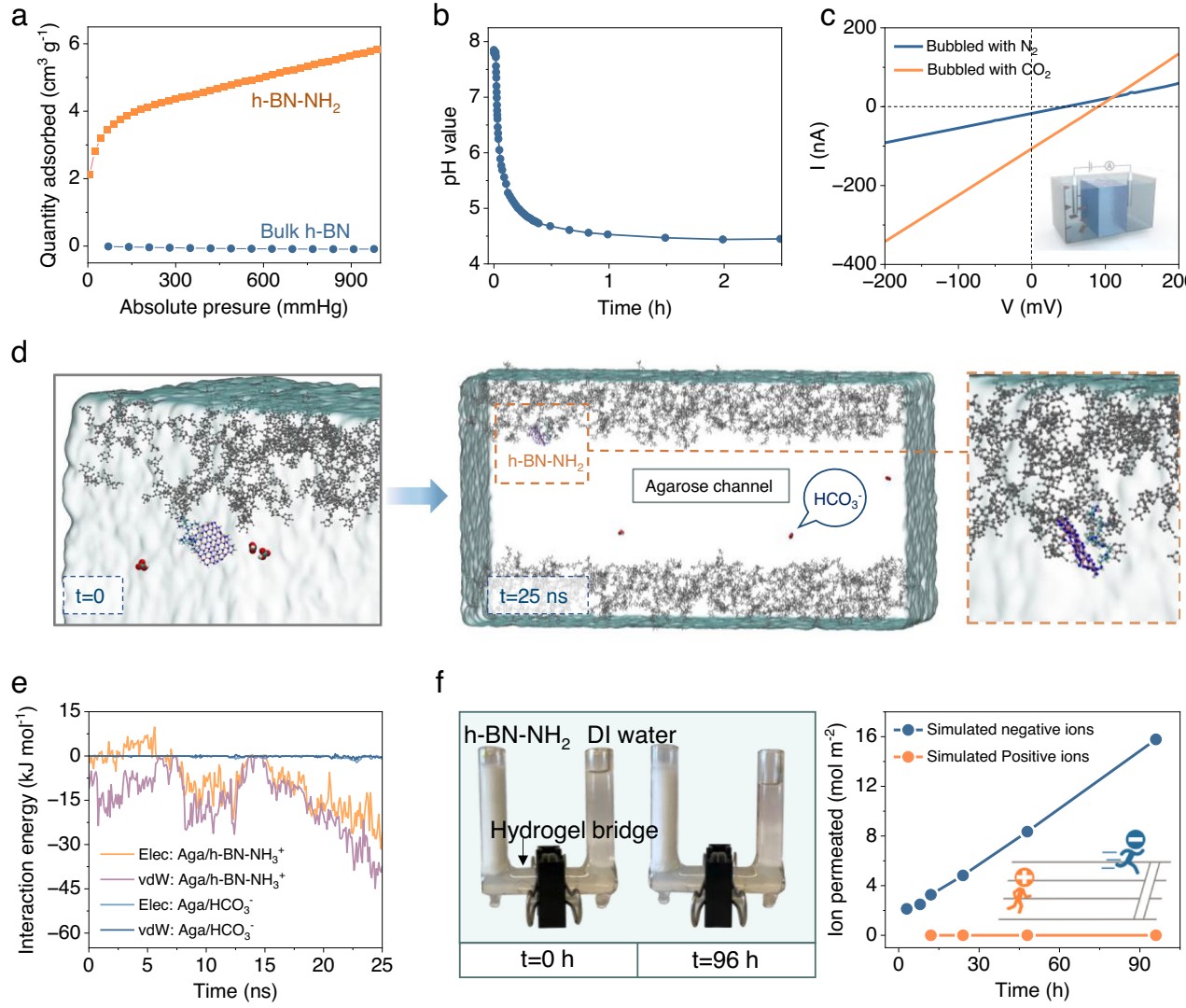

**Fig. 3 | Mechanism of the electricity generation from CO₂ adsorption.**
**a** Comparison of the adsorption isotherms (293.15 K) of $CO_2$ by h-BN-NH₂ nanosheets and bulk h-BN measured by Brunauer–Emmett–Teller (BET) analysis.
**b** The pH value of h-BN-NH₂ water solution as a function of $CO_2$ adsorption time, which drops gradually because of the generation of $HCO_3^-$ ions. Total solution: 100-mL water solution of h-BN-NH₂ nanosheets at 1 mg mL⁻¹. $CO_2$ feeding rate is 10 mL min⁻¹. **c** $I–V$ curves of h-BN-NH₂ nanosheet water solution bubbled with $CO_2$ and N₂ respectively in an H-cell. The other side was fed with DI water. **d** MD simulation model of agarose channel for selective transport of h-BN-NH₃⁺ and bicarbonate ions. The channel size was set to be wide enough to eliminate size effects. The relative positions of h-BN-NH₃⁺ and $HCO_3^-$ were investigated at t = 0 and t = 25 ns respectively, which shows that h-BN-NH₃⁺ get anchored while $HCO_3^-$ freely move within and across the channel. **e** The electrostatic and van der Waals inter-action energies between agarose and $HCO_3^-$ compared with those of between agarose and h-BN-NH₃⁺ nanosheet. **f** Ion permeation rate comparison of negative ions ($HCO_3^-$) and positive ions (h-BN-NH₃⁺) under concentration gradient. A homemade diffusion setup was applied to confirm the simulation results with a hydrogel bridge in between two reservoirs containing h-BN-NH₂ and DI water. (further experimental details are provided in Supplementary Note 4).

hydrogel matrix is 102 nm on average, which is much larger than the physical size and the hydrated radius ($\lambda$ ~ 4 Å) of $HCO_3^-$. The size dif-ference significantly reduces the physical and charge effects and low-ers the energy barrier for $HCO_3^-$ movements in the ion channel. Notably, the hydrogel channels greatly limit the h-BN-NH₃⁺ ion diffu-sion due to two main reasons. First, the lateral size of the h-BN-NH₃⁺ nanosheets (100–300 nm) is comparable to the hydrogel pore size, as a result, these nanosheets are physically trapped within the network of hydrogel chains. Second, the -OH groups on agarose chains can form strong interactions with h-BN-NH₃⁺ nanosheets through Van der Waals force and electrostatic interaction. Computational calculation indi-cates that the interaction energy of agarose chains with h-BN-NH₃⁺ nanosheets is up to ~80 KJ mol⁻¹, while that of bicarbonate ions only peaks at ~2.2 KJ mol⁻¹ (Fig. 3d, e, Supplementary Fig. 11). These results show that the confined size exclusion effect, combined with intensive

molecular interactions of the interlocked 3D hydrogen bond in hydrogel, anchors the nanosheet ions and severely limits their move-ment within the matrix.

To gain further insight into selective ion transport within the hydrogel, an ion-diffusion experiment was conducted. Two cells of a custom-made diffusion device, acting as ion-releasing/receiving com-partments, were separated by a hydrogel bridge serving as the selec-tive ion channel. A h-BN-NH₂ nanosheet water solution and a sodium bicarbonate salt solution, representing positive and negative ions, respectively, were separately filled into the feeding cell while the receiving cell was filled with deionized (DI) water (Supplementary Fig. 12). The concentration gradient between the two cells drove the h-BN-NH₂ nanosheets or sodium bicarbonate ions to diffuse from their respective cells to the DI-water cell through the hydrogel bridge. Quantitative analysis revealed a high diffusion rate of sodium

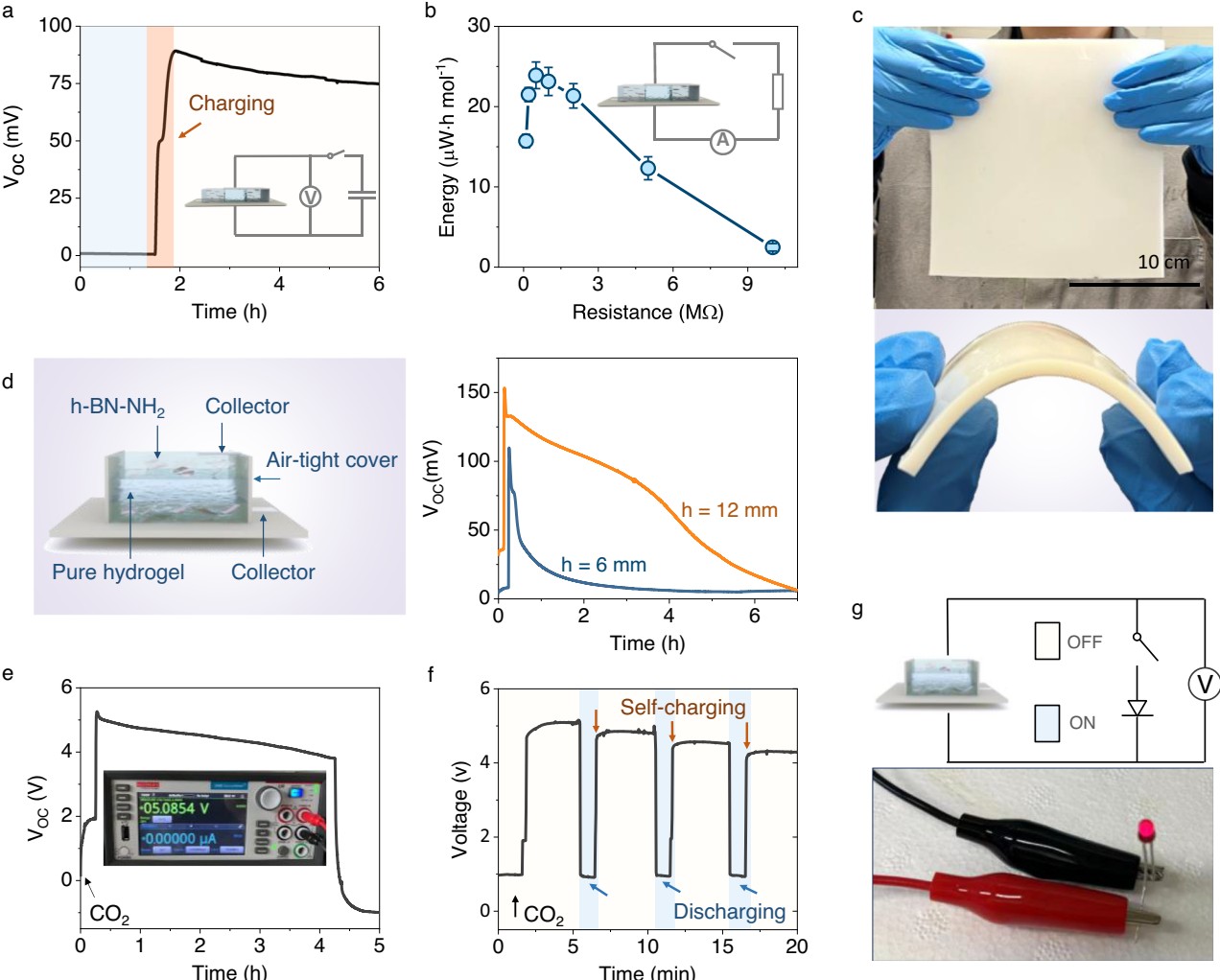

**Fig. 4 | Scaling up and practical demonstration $CO_2$ adsorption electricity generator. a** $CO_2$ adsorption induced electricity was stored by charging a commercial capacitor (0.5 µF). Blue, yellow background indicate the potential of the capacitor at initial and charged status, respectively. **b** Energy harvested from the generator in a one-adsorption cycle when connected with different external resistors. The total electricity was calculated according to the current curve as a function of time monitored by the source meter: $W = \int RI^2(t)d(t)$. Error bars are standard deviations from three tests. **c** Photographs of a large size of NAH composite fabricated by casting method depicting scalability. **d** Schematic of the vertically staked generator and the power generation from the devices with different thicknesses (h indicates the total thickness of the devices). By using the vertically stacked generator, the accelerated ion transport leads to a prompt response to the

$CO_2$. The induced $V_{oc}$ of the generators reached their peaks in several seconds after feeding $CO_2$ into the testing box. In addition, the thicker the generator, the higher the peak $V_{oc}$, and the longer the electrical signals last. **e** Electricity generation by five parallel connection of vertically stacked generator groups comprising ten generators in series (5 × 10). Inset is the photograph of the output voltage measured by the source meter. **f** Generators in parallel and series (5 × 10) were used to power a light-emitting diode, the blue and red background indicate the status of the switch (on and off). The fluctuations on the *Voltage* ascending curve likely indicate that the charging speed is influenced by the characteristics of the external circuit. **g** Photograph of the emitting diode (Broadcom Inc. Bulb size: 3 mm, forward voltage: 1.6 V, forward current 1 mA) powered by the integrated $CO_2$ generators.

bicarbonate ions in the hydrogel at 0.16 mol m$^{-2}$ h$^{-1}$, which is roughly $1.8 \times 10^6$ times faster than that of h-BN-NH$_2$ nanosheets in a testing span of four days ($D_-/D_+$, Fig. 3f, Table S2, and Note S4). We also monitored the $I–V$ curves of h-BN-NH$_2$ water solutions after $CO_2$ adsorption in an H-cell, with a 3-fold concentration gradient separated by pure hydrogel. From this, a transference number of anion (t-) nearing 1 was derived. This outcome further indicates the near-perfect anion/cation selectivity achieved by the hydrogel channels (Supplementary Fig. 13, Note 5). Therefore, the distinguishable ion diffusion rate in artificial channels between oppositely charged ions is the underlying principle[33,34] for the ion-transport-induced electricity generation process in our NAH electricity generator.

Experimental verification was also conducted by controlling the length of the ion diffusion path, specifically by adjusting the width of the pure hydrogel in the middle (Supplementary Fig. 14). Reducing the

width from 3 cm to 1 cm increased the peak $I_{SC}$, escalating from 140 nA to 169.7 nA, indicating a higher peak ion diffusion flux. Moreover, it achieved the peak $I_{SC}$ in only 11.4 min compared to the 20.4 min required by the prototype of 3 cm. Additionally, the time taken to decay to half its peak $I_{SC}$ decreased to 24.6 min from 40.2 min. This expedited charging and discharging process, achieved through the reduction of the diffusion path, agrees well with the characteristics of ion-diffusion-induced power generation.

**Practical demonstration of NAH electricity generator**

The practicality of NAH electricity generator was evaluated in terms of energy storage, conversion efficiency, scalability, response time, and modular integration. The generated electricity can be stored by charging a commercial capacitor (0.5 µF) (Fig. 4a). When connected to an external circuit with a 0.5 MΩ resistor during an adsorption cycle, the

generator reaches its peak total energy generation of 24 μW·h mol$^{-1}$ with a volumetric density of 0.0027 μW cm$^{-3}$ (Fig. 4b), corresponding to an energy conversion efficiency of approximately 0.6% relative to the total input energy (Supplementary Notes 6, 7 and Supplementary Fig. 15). Although the energy conversion efficiency aligns with those of recently reported moisture electricity generators, the present peak density falls short of the required threshold[20,24]. Given that near-perfect selectivity has been achieved, the bottleneck issue affecting the current low power density appears to be the insufficient ion flux in the system. Therefore, to improve the power density of the NAH power generator, we pursued two distinct strategies: optimizing the device configuration for improved ion transport efficiency and enhancing $CO_2$ adsorption capacity for a higher concentration gradient. These two approaches achieved an overall 10-fold improvement in the power density. Future research is encouraged to focus on boosting the ion flux while maintaining the high selectivity to further increase efficiency and thus consolidate the practical implications (Supplementary Fig. 16, Supplementary Table 3.).

To demonstrate the scalability of NAH electricity generator, a 180 mm × 180 mm × 4 mm NAH composite hydrogel was fabricated. The hydrogel composite film was then cut into small generators (60 mm × 20 mm × 4 mm). Connecting four of them in either series or parallel, led to a linear increase in output $V_{OC}$ of approximately 280 mV or $I_{SC}$ of approximately 400 nA (Fig. 4c, Supplementary Fig. 17). As the electricity generation process of NAH electricity generator is primarily determined by selective ion transport within NAH hydrogel matrix, the speed of electricity generation can be theoretically maximized by rational designing the ion-diffusion path length and the effective ion-diffusion area. To prove this hypothesis, we designed another type of the NAH electricity generator, in which two NAH hydrogels (20 mm × 20 mm × 4 mm) sandwiched a pure hydrogel of the same size but were vertically stacked (Fig. 4d). This configuration reduces the length of the ion channel while expanding its effective area (Supplementary Fig. 18). A prompt rather than time-lagged electricity generation process was observed in the vertical configuration, reducing the response time of NAH electricity generator from minutes to seconds (Fig. 4d).

When 50 generators were integrated with the vertical configuration, a stable output voltage of 5 V was generated (Fig. 4e). The generated electricity was sufficient to power a light-emitting diode (Fig. 4g. with an operating voltage of 1.6 V and forward current of 1 mA, *watch* Supplementary Movie 1). Upon lighting the diode, the induced voltage dropped below the operating voltage but was rapidly recovered once the external circuit was disconnected. Only a minor reduction in the voltage was observed after four cycles of charge and discharge (Fig. 4f). This discharge and self-charge behavior further confirm that the electricity generated by the device results from selective ion transport. The voltage is built by ion relocation driven by a chemical potential gradient. Unlike rapid electrode redox reactions in chemical batteries, the ion-transport-induced electricity generators take time to accomplish the ion-releasing and selective transport to build the potential difference.

In summary, we have successfully developed NAH-based generators that are able to generate electricity from $CO_2$ adsorption. The developed generators take advantage of $CO_2$ adsorption-induced ion releasing and ion separation within hydrogel channels to facilitate power generation. Our approaches to engineer ions, encompassing controlling their physical size and intramolecular interactions, demonstrate remarkable efficiency for achieving sharp ion separation across the specifically designed channels. The as-established strategies are applicable to other diverse ion separation involved purification, energy harvesting and resource recovery processes. We believe that the establishment of $CO_2$-adsorption energy harvesting framework could catalyze the development of alternative carbon-negative power generation solutions, thereby enhancing the sustainability of existing

CCUS technologies by simultaneous carbon capture and energy conversion.

## Methods

### Synthesis of the PEI functionalized nanosheets

The PEI-functionalized nanosheets were synthesized using a sticky exfoliation process as previously described in our work[29]. Briefly, 0.5-g pristine bulk layered materials such as hexagonal boron nitride (powder, ~1 μm, 98%, Sigma-Aldrich), graphite (Flakes, 99% Carbon, Sigma-Aldrich), carbon nitride (Supplementary Note 2) and 2 g PEI (Mn at ~10,000 by GPC, average Mw at ~25,000 by LS, Sigma-Aldrich, 408727) were charged into a 250-mL $ZrO_2$ milling jar. Three types of $ZrO_2$ balls with different weights and diameters (100 g, d = 10 mm, 200 g, d = 5 mm and 20 g, d = 0.1 mm) were used as grinding balls. The milling jar was loaded into a planetary ball mill (ZQM-P2, Changsha Mitrcn Instrument Equipment Co., Ltd, Revolution radius: 10 cm, Rotation radius: 39 mm), and the rotation speed was set at 250 rpm for revolution and 500 rpm for rotation. After milling for a set duration (10 h unless stated otherwise), 100 g of DI water was added to the milling jar to wash out the nanosheets and PEI mixture. The resulting water solution containing the mixture was filtered onto nylon membranes (pore size: 0.45 μm, diameter 47 mm, Sterlitech, USA) and rinsed with water repeatedly until the pH value of the outlet water reached 8. The nylon membranes with the nanosheets atop were then subjected to ultrasonication for 30 min (Unisonics FXP12M, 40 kHz, 100 W) to re-disperse the nanosheets in water. Finally, the dispersion was centrifuged at RCF of 236 g (Rotor 12181, Sigma 2-16 P) for 20 min to remove thick flakes. The supernatant was decanted as the final water dispersion product of PEI-functionalized nanosheets.

### Fabrication of the generators

Agarose powder (Sigma-Aldrich, BioReagent, gel point 36 °C ± 1.5 °C) was dissolved in water at 80 °C to obtain the hydrogel precursor solutions at concentrations of 2 wt% and 4 wt%, receptivity, and were stored at 70 °C oven for generator fabrication. A PDMS (Sylgard 184, Dow Corning) rubber cell measuring 6 mm × 2 mm × 0.4 mm was cut and used as the generator mold. Current collectors were created by coating silver paint at the bottom of the two ends of the PDMS cell. Carbon tape and platinized titanium mesh (Fuel cell store, 592770) were also adopted as current collectors. After embedding electric collectors, pure agarose solution (2 wt%) at 70 °C was firstly drop cast into the PDMS mold and left in the atmosphere for 30 min to undergo the sol-gel transition to form a hydrogel film. Two ends of the pure hydrogel film (2 mm × 2 mm × 0.4 mm) were then graved with only middle part left in the mold. Second, the nanosheets solution with a concentration of 2 mg mL$^{-1}$ (unless stated otherwise) was heated to 70 °C and mixed with equal volume of agarose precursor solution at the concentration of 4 wt%. The obtained mix solution was drop casted into the two ends of the PDMS mold and left for another 30 min to finally obtain the 3-module NAH generator. For the whole piece of NAH generator, the agarose and nanosheet mix solution was directly drop casted into the corresponding mold without the procedure of casting pure agarose solution. The device was subsequently connected to an external circuit equipped with a source meter (Keithley 2450) to monitor the generator's electrical signals during testing.

### Electrical measurements

The open circuit voltage and short circuit current of the NAH composite generator were measured using a source meter (Keithley 2450) under ambient conditions. The current output test was set to 0 nA and the voltage was set to 0 V for the open circuit and short circuit measurements, respectively. The generator was placed in a testing box (50 cm by 25 cm) with a $CO_2$ inlet at its bottom and outlet at the top, and the two collectors on the left/right sides were short connected via an external circuit to discharge the device until the $I_{SC}$ dropped below

5 nA. This pre-discharging process helped to reduce the influence of slight chemical potential differences within the hydrogel composite caused by artificially introduced asymmetric structure. $CO_2$ was then fed into the box at a flow rate of 2.5 L min$^{-1}$ for 3 min, after which the device was covered for further testing. To evaluate the power density, the generator was connected with external resistors of varying resistances, and the external current was monitored for density calculation. The I-V curves of the nanosheet water solution were obtained using a pair of Ag/AgCl electrodes with applied voltages ranging from −0.2 V to 0.2 V.

## Generator regeneration

The regeneration of the generator was explored by choosing a pH-swing strategy to study its multiple utilization potential[25]. Following 5 cycles of electricity generation, the generator was removed and immersed in a calcium hydroxide (Ca(OH)$_2$, > 95%, Sigma-Aldrich,) solution with a pH value of 10 for 1 h. This was followed by washing the generator in excessive DI water 3 times. Subsequently, the generator was placed back in the testing box and subjected to the same testing procedures.

## Material characterizations

The network structure of the agarose hydrogel and the composite generator was analyzed using various methods. The network structure of the agarose hydrogel and the composite generator was studied by a scanning electron microscope (SEM) (Nova NanoSEM 450, FEI, U.S.A.) operated at 3 kV with a working distance of 5 mm. To prepare samples, the hydrogel was first immersed in liquid nitrogen to quickly freeze the structure and then followed by freeze-drying (FreeZone 2.5 liters, Labconco Corporation, USA). The obtained samples were coated with iridium (1.5 to 2 nm thick) prior to SEM characterization. The elemental content (C, N, and H) of samples was carried out on an Elemental Analyzer (FlashSmart, Thermo Scientific) using powder samples (freeze-dried from the nanosheet water solution). X-ray photoelectron spectroscopy (XPS) was performed using h-BN powder samples on a Thermo Scientific Nexsa Surface Analysis System equipped with a hemispherical analyzer. X-ray diffraction (XRD) analysis was performed from 5 to 100 °C by a diffractometer equipped with Cu Kα radiation (Miniflex 600 X-ray diffractometer, Rigaku, Japan; D2 PHASER powder diffractometer, Bruker, Germany) using powder bulk h-BN and h-BN nanosheet sample. TGA was performed in the range of 50 to 900 °C with the temperature rising rate of 10 °C min$^{-1}$ under continuous nitrogen flow (PerkinElmer TGA 8000).

## Molecular Dynamics (MD) simulation study

MD simulation was applied to compare the transport behavior of $HCO_3^-$ ions and h-BN-$NH_3^+$ through the agarose channel with a channel size of 8 nm. The systems were filled with water through the solvation process. The water molecules were described by the SPC/E model[35]. The universal force field UFF forcefield[36] with QEq charge is assigned to h-BN-$NH_3^+$ [37] and other molecules through Open Babel[38] and OBGMX[39] codes. The BN sheet was frozen in the simulations during the simulation process. Systems were subjected to multi-step steepest-descent energy minimization followed by CG energy minimization. After that, 100 ps NVT equilibrium simulations and 100 ps NPT equilibrium simulations were performed. The time step is 2.0 fs, and bonds to hydrogen atoms were maintained with the LINCS algorithm[40]. And a constant simulation temperature of 298.15 K was maintained by the V-rescale thermostat, the pressure coupling was reached through Parrinello-Rahman[41]. The rcoulomb and rvdw were set as 10 Å. The electrostatic interactions were evaluated using the particle mesh Ewald algorithm. After that, a 50 ns production simulation was performed. All MD simulations in this work were performed using the GROMACS 2019.6[42]. Simulation results were analyzed and produced using VMD software (VMD for WIN64, version 1.9.3, September 6, 2020)[43].

## Data availability

All data in this study are included in this article and its supplementary information file. Raw data can be obtained from the author upon request. Source data are provided with this paper.

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

## Acknowledgements
We thank Yun Xia, Zhonghao Xu, and Xin Zeng for designing and drawing graphics. We acknowledge the financial support from the ARC Centre of Excellence for Green Electrochemical Transformation of Carbon Dioxide (CE230100017), and the ARC Industry Transformation Research Hub for Energy-efficient Separation (IH170100009). X.W.Z. thanks the Australian Research Council for his ARC Future Fellowship (FT210100593). This work made use of the facilities at the Monash Centre for Electron Microscopy (MCEM) and Melbourne Centre for Nanofabrication (MCN).

## Author contributions
Conceptualization was done by Z.Y.W., H.T.W., and X.W.Z. The methodology was developed by Z.Y.W. and X.W.Z. Characterizations were carried out by Z.Y.W., X.K.Z., X.F.L., F.D., and H.Z. Computational study and simulations were designed and performed by T.H. Data analysis and validation were carried out by M.T., Z.Y.W., and Y.K. All authors have read and agreed to the published version of the paper.

## Competing interests
The authors declare no competing interests.
