## [Peer Review File · Nature Communications]

Electricity Generation from Carbon Dioxide Adsorption by Spatially Nanoconfined Ion SeparationREVIEWER COMMENTS

Reviewer #1 (Remarks to the Author):

In this work agarose based hydrogels containing poly(ethylene imine) (PEI) functionalized nanosheets as fillers are used as absorber for CO₂. The authors showed that these hydrogels can be used to generate electricity owing to the selective diffusion of HCO₃⁻ ions. The main concept of materials design used in this work is- the amine group containing PEI is grafted to the nanosheets which prevent the diffusion of -NH₃⁺ moieties, while the counterion HCO₃⁻ can diffuse freely through the water filled channel. I must admit that the authors put a lot of efforts for the proof of principle. However, looking at the overall potential impact of the work for practical power generation I do not think it is a breakthrough like the authors claimed in the manuscript. The concept of using selective ion diffusion through a semipermeable barrier to generate electricity is not new. The authors have also pointed it out in the introduction of the paper. It is true that hydrogels capable of adsorbing CO₂ are not commonly used for this purpose. The authors have shown that by stacking 4 electricity generators containing 60mm×20mm×4mm sized hydrogels VOC of approximately 280 mV or ISC of approximately 400 nA was achieved. It is important to determine the maximum power density (i.e. maximum output power per unit area of the hydrogels) of these hydrogels and compare with that obtainable by other techniques e.g. salinity gradient. In this case the concentration of HCO₃⁻ generated per unit area of the hydrogels are rather low and the concentration difference creates the driving force of HCO₃⁻ ion diffusion between the hydrogel containing filler and hydrogel without filler. Therefore, the overall achievable maximum power from per unit area of the hydrogels will not be of any practical use. The hydrogel design principle itself largely limits the -NH₂ group content per unit area.

The paper does not make any contribution to the fundamental understanding of electricity generation owing to selective ion diffusion. It only reports, if the PEI functionalized 2D nanosheets are used as fillers in the hydrogels, these hydrogels can take advantage of the diffusion of HCO₃⁻ for generation of electricity. In my opinion that is not sufficient for publication in "Nature Communications". I do not think it makes sufficient contribution which will lead to future studies to progress the power generation using this type of PEI functionalized nanosheet filled hydrogels which can absorb CO₂. Although overall it is a good paper which will be suitable for other journals, I do not recommend publication in "Nature Communications".

Reviewer #2 (Remarks to the Author):

Carbon dioxide capture and utilization is one of the key issues toward the sustainable development of society. This manuscript focused on carbon dioxide adsorption-based electricity generation, and presented a quite unique strategy for selective ion transport, by spatially nanoconfined separation of HCO₃⁻ anions and h-BN nanosheet-NH₃⁺ cations with huge size differences, to achieve efficient power generation. Different from ion-exchange membranes, neutral nanoporous hydrogel matrix was used to

spatially hinder h-BN nanosheet-NH₃⁺ movement while allow HCO₃⁻ transport freely. In general, the manuscript was well organized and the idea was fairly verified by experiments and calculations. The following major points need to be properly addressed ahead of publication.

1. Is the sandwiched configuration necessary for the NAH electricity generator? I understand that the pure agarose hydrogel section is used for ion selection and separation. However, the role of the NAH hydrogel at the ion receiving reservoir side seems unclear. What happens if it is replaced by water?
2. What is the influence of the thickness of the pure agarose hydrogel on the ion transport rate and the power density? Usually, shorter diffusion path of ions facilitates higher ion flux and power density. It is recommended that the authors need to clarify this by experimental verifications.
3. What is the maximum or optimal concentration of h-BN-NH₂ nanosheets that can be used for a best power generation performance?
4. There are some language mistakes in the manuscript that need to be corrected thoroughly. For example, in Figure 3f, the unit of vertical axis.
5. During the pH swing process, can the CaCO₃ precipitant be removed completely from the hydrogel matrix? Otherwise, will it block the nanopores of the matrix and hinder the performance of the generator?
6. Generating electricity from the adsorption of CO₂ has also been reported in recent works (e.g., <https://doi.org/10.1016/j.greenca.2023.08.002>). For a comprehensive literature review, these works should be cited and their contribution to this field should be properly acknowledged.
7. Although the NAH electricity generator works, its current and the output power density is quite low, which may hinder its practical applications. Is there any way to improve the energy conversion efficiency? I encourage the authors to provide a suitable perspective and discussion in the conclusion part.

Reviewer #3 (Remarks to the Author):

This is an interesting study of electricity generation from CO₂ absorption in an agarose hydrogel. The key idea is that the mobilities of the anion and cations generated by CO₂ absorption can be differentiated by coating the CO₂-absorbing PEI polymer onto nanoplatelets like BN or graphene sheets. The cationic species generated by the CO₂ absorption is then bound to the relatively large nanoplatelet (e.g., BN or graphene sheets) while the smaller anion is free to diffuse through the hydrogel. This is a novel idea, to my best knowledge. The authors convincingly demonstrate that the nanogenerators do produce electrical power as a result of the difference in mobilities of the ions through the hydrogel. I find the paper to be interesting, well written, and a potentially valuable contribution to the literature on energy conversion and CO₂ capture. I do have the following comments and suggestions:

- 1) Some further analysis of the electricity generation would be instructive, as the open-circuit osmotic voltage across the hydrogel can be directly related to the ion selectivity. Specifically, the transference number t_+ or the ratio of cation-to-total-ion flux is related to the V_{osm} as:

$$t_+ = J_+ / (J_+ + J_-) = (1/2) \{V_{osm} / [(RT/zF \ln((\gamma_H C_H) / (\gamma_L C_L)))] + 1\}$$

where $\gamma_{H,L}$, R , T , z , F , are the activity coefficient, ion concentration, gas constant, temperature, valence, and Faraday constant, respectively. The subscript H and L would refer to the high- and low- ionic concentration sides across the hydrogel. For this particular study, one could assume that the mobility of the cation is negligible, so that $t_+ \rightarrow 0$. It seems then that the open circuit osmotic voltage would depend approximately as $\ln(C_H/C_L)$. Could the authors use the data shown in Fig. 2c&d to test this? By increasing the concentration of h-BN-NH₂ or the PEI grafting ratio, they would essentially be increasing C_H . In my back-of-the-envelope estimation, going from an h-BN-NH₂ concentration of 1 to 5 mg/ml would lead to an expected $\ln(5)=1.6x$ increase in open-circuit voltage. This is pretty close to what is seen from the measured data in Fig. 2c. However, the increase in PEI grafting ratio shown in Fig. 2d seems to give a greater-than-expected increase in V_{osm} . Could the authors comment on why increasing the PEI grafting ratio seems to increase the electricity generation more than both theory and increasing h-BN-NH₂ concentration?

2) The I-V data in Fig. 3c is puzzling. Why do the I-V curves for the h-BN-NH₂ solution not go through the origin BEFORE CO₂ absorption? In fact, the increase in V_{oc} and I_{sc} after CO₂ absorption is relatively small compared to what exists beforehand. That would seem to imply that the hydrogel was ion selective and generating significant electrical power even before CO₂ absorption. Could the authors comment on this?

3) Were the redox potentials at the electrodes subtracted from the measured potentials to get the true V_{osm} ?

4) It would be helpful to estimate the areal or volumetric power density of the generator. While I suspect that it would be low compared to other technologies like conventional reverse-electrodialysis membranes, the power density would still be a useful benchmark for future studies of CO₂ absorbing nanogenerators. On a related note, I do not understand the units on the "energy density" shown in Fig. 4b. Is it really "microW hr mol"?

5) Why are there "glitches" as V_{oc} rises initially in Figs 4a, e, f? This may also appear in the data shown in Fig. 2a, although it is harder to see.

Some minor points:

1) In Fig. 2b, how long was the regeneration cycle? Can it be pointed out in the data? Is it perhaps the small bump before $t=50$ hr? And was it done with the generator in situ, or was the hydrogel removed, soaked, and then replaced? Some clarification here would be appreciated.

2) V_{oc} and I_{sc} are mis-labeled in Fig 3c.

3) What is " λ_D " of HCO₃⁻ on page 10?

4) The diffusion experiments discussed on pg. 10 are a little unclear. Was the hydrostatic difference

that built up due to water transport considered or can it be argued to be negligible? In Fig. 3f, the image on the right shows that the column of water on the h-BN-NH₂ side increased in height over the course of 96 hours. Also, how was the concentration measured for the sodium bicarbonate experiments? Was the diffusion of Na⁺ accounted for? When the authors say that the diffusion of sodium bicarbonate was 1.8×10^6 times faster than that of h-BN-NH₂, are they using the transport rate of HCO₃⁻ only, or of both Na⁺ and HCO₃⁻? It would seem fairer to compare only the rate of HCO₃⁻ to that of h-BN-NH₂.

5) The units of the interaction energies of the various ions with the agarose chains seem to be off. Are they supposed to be KJ/mol rather than KJs as currently given in the manuscript?

Response Letter

Reviewer #1 (Remarks to the Author):

General comments Part A: In this work agarose based hydrogels containing poly(ethylene imine) (PEI) functionalized nanosheets as fillers are used as absorber for CO₂. The authors showed that these hydrogels can be used to generate electricity owing to the selective diffusion of HCO₃⁻ ions. The main concept of materials design used in this work is- the amine group containing PEI is grafted to the nanosheets which prevent the diffusion of -NH₃⁺ moieties, while the counterion HCO₃⁻ can diffuse freely through the water filled channel. I must admit that the authors put a lot of efforts for the proof of principle. However, looking at the overall potential impact of the work for practical power generation I do not think it is a breakthrough like the authors claimed in the manuscript. The concept of using selective ion diffusion through a semipermeable barrier to generate electricity is not new. The authors have also pointed it out in the introduction of the paper. It is true that hydrogels capable of adsorbing CO₂ are not commonly used for this purpose.

The paper does not make any contribution to the fundamental understanding of electricity generation owing to selective ion diffusion. It only reports, if the PEI functionalized 2D nanosheets are used as fillers in the hydrogels, these hydrogels can take advantage of the diffusion of HCO₃⁻ for generation of electricity.

Authors: We appreciate the reviewer's recognition of our efforts in this study. However, we respectfully differ in opinion regarding the reviewer's assessment of its scientific contributions. As outlined in the introduction of our manuscript, the key contributions of our work are summarized below, which extend the potential and applicability of ion-diffusion-induced energy generation technologies.

1. New strategy for achieving high throughput anion/cation selective transportation.

Charge-selective transportation is vital for various processes like energy conversion, storage, sensing, and signal propagation. Previous methods for charge separation relied on electrical or size effects. Electrical effect leverages the overlapped electrical double layer (EDL) within charged ion channels. It is effective when the channel's radius is smaller than the Debye length

of transported counter ions. Size effect involves precise control of channel sizes within the Å to nm scale, ensuring the channel size aligns between the sizes of the ions to be separated. Reviewing recent works on charge separation highlighted persistent challenges of struggling to achieve efficient separation at a bulk scale (Table R1). Rather than solely relying on controlling ion channels, we showcased that **cascading charge separation can be achieved by manipulating the size of the ions to be separated**. With a substantial size difference of over two magnitudes, nanosheet cations remain firmly fixed by the hydrogel matrix while anions can freely pass through its water-filling channels. This led to achieving a charge selectivity ratio (D^-/D^+) as high as 1.8×10^6 , which exceeds most of the existing data reported in the literature, except for the extraordinary charge selectivity previously realized within a nanoscale single carbon nanotube.

Table R1. The reported charge selectivity achieved by semi-barriers.

Charge separation	selectivity	Strategy	Reference
Anion/cation	~2.5	Positively charged cellulose membrane	1
Cation/anion	9	Negatively charged graphene oxide membrane	2
Cation/anion	29	Cation-selective two-dimensional polyimine membranes	3
Cation/anion	1.3-2.2	Aramid nanofibers and BN nanosheets mixed matrix membrane	4
Cation/anion	2.3-4	Silk-Crosslinked Membranes	5
Cation/anion	1.6-19	Subnanochannelled membranes by metal-organic frameworks	6
Cation/anion	∞	0.8nm carbon nanotube, channel charge and size regulation	7
Anion/cation	1.8×10^6	Targeted ion size manipulation	This work

Through the new strategy on ion size regulation, we demonstrated **a possible way of achieving precise charge separation in a high-throughput manner**. This method presents a more scalable approach than constructing narrow distributed channels for ion separation. To

reinforce this claim, we conducted an additional experiment to calculate the transference numbers of anions and cations. The obtained results indicated that a transference number of anions (t_-) approaching 1 ± 0.05 was achieved in our study, further supporting the near-perfect anion/cation selectivity. Please refer to **the response to question 1 of reviewer 3** for the details of the testing and calculation.

2. Using problematic greenhouse gases for energy generation.

In addition to aiming for high energy density or efficiency, it is equally promising to harvest energy from problematic sources. Considering the example of osmotic energy harvesters, regardless of their efficiency and energy density, their feasibility can encounter challenges in regions that are facing freshwater shortages⁸. In contrast, carbon dioxide (CO₂) represents a widespread waste gas with an urgent need for effective capture. The concept of generating electricity through its adsorption process introduces a compelling incentive, positioning it as a promising and inspirational area of exploration for researchers in both the energy generation and carbon capture domains. In a similar vein, this approach could set the stage for harnessing energy from other challenging waste gases, such as sulfur dioxide (SO₂) and nitrogen dioxide (NO₂). Utilizing these waste gases for energy production has the potential to simultaneously tackle multiple environmental issues, fostering paths towards sustainable energy solutions and reducing harmful emissions.

We have thoroughly enhanced our discussion of the two contributions in the revised manuscript, with a special focus on enriching the content in the introduction section. Some revisions are exemplified as follows. Kindly refer to more details in the revised introduction and conclusion section.

However, the positively charged amine ions and negatively charged bicarbonate ions generated in the conventional CO₂ adsorption process possess similar sizes at the molecular level and are well mixed, making their efficient separation challenging^{10,27,28}. Dependence solely on a positively charged separation membrane yielded a mobility selectivity of 2.5 between HCO₃⁻ and R-NH₃⁺ ions²⁶, a significantly lower magnitude compared to what was attained in the osmotic energy generation process¹⁹. Addressing this challenge could potentially facilitate the pursuit of efficient energy conversion and sufficient power density in this promising field.
(Main text Page 2)

To overcome the size similarity of the generated ions by traditional adsorbents, nanocomposite adsorbents are synthesized by grafting polyethyleneimine (PEI) molecules with abundant -NH/-NH₂ groups for CO₂ adsorption onto 2D hexagonal boron nitride (h-BN) nanosheets, forming PEI-functionalized h-BN nanosheets (h-BN-NH₂) (Fig. 1b i, Supplementary Fig. S1).

(Main text Page 3)

Since the extended h-BN skeleton of the positive ions is trapped by the agarose hydrogel matrix while the negative HCO₃⁻ ions are left freely in the interconnected water-filled channels, these oppositely charged ions diffuse across the generator at a remarkable rate difference (>10⁶). (Fig. 1b iii). Such a diffusion rate gap is translated into electricity and further amplified to power external electronic devices. This study utilizes the gaseous CO₂ as an energy-harvesting target. The demonstrated ion engineering strategies could advance other ion-separation-related energy, resource, and environmental applications. **(Main text Page 3)**

Quantitative analysis revealed a high diffusion rate of sodium bicarbonate ions in the hydrogel at 0.16 mol/m²·h, which is roughly 1.8×10⁶ times faster than that of h-BN-NH₂ nanosheets in a testing span of four days (D/D_+ , Fig. 3f, Table S2, and Note S4). We also monitored the I - V curves of h-BN-NH₂ water solutions after CO₂ adsorption in an H-cell, with a 3-fold concentration gradient separated by pure hydrogel. From this, a transference number of anion (t_-) nearing 1 was derived. This outcome further indicates the near-perfect anion/cation selectivity achieved by the hydrogel channels (Supplementary Fig. S13, Note S5). **(Main text Page 10)**

General comments Part B: The authors have shown that by stacking 4 electricity generators containing 60mm×20mm×4mm sized hydrogels V_{OC} of approximately 280 mV or I_{SC} of approximately 400 nA was achieved. It is important to determine the maximum power density (i.e. maximum output power per unit area of the hydrogels) of these hydrogels and compare with that obtainable by other techniques e.g. salinity gradient. In this case the concentration of HCO₃⁻ generated per unit area of the hydrogels are rather low and the concentration difference creates the driving force of HCO₃⁻ ion diffusion between the hydrogel containing filler and hydrogel without filler. Therefore, the overall achievable maximum power from per unit area

of the hydrogels will not be of any practical use. The hydrogel design principle itself largely limits the -NH₂ group content per unit area.

In my opinion that is not sufficient for publication in "Nature Communications". I do not think it makes sufficient contribution which will lead to future studies to progress the power generation using this type of PEI functionalized nanosheet filled hydrogels which can absorb CO₂. Although overall it is a good paper which will be suitable for a other journals, I do not recommend publication in "Nature Communications".

Authors: We again thank the reviewer for acknowledging the overall quality of the paper. The maximum volumetric power density has been calculated and provided in the revised manuscript as requested (**Main text page 11**). Please refer to **Supplementary Note S7** for the detailed calculation.

We agree with the reviewer that the power density of our electricity generator is still low. However, it's important to note that comparing an early-stage innovation like ours to well-established technologies may not provide an entirely fair assessment. We firmly believe that our electricity generator holds significant potential for further improvements. Theoretically, the power density ($I_{\text{dif}} \times P$) of our electricity generator is linked to the ion selectivity (D_1-D_2) and ion flux ($qD_2A \frac{dC}{dx}$) as shown in the two equations.

$$I_{\text{dif}} = - (qD_1A \frac{dC_1}{dx} - qD_2A \frac{dC_2}{dx})$$

$$P = \frac{qA(D_1-D_2)}{\sigma} \int \frac{dC_0}{dx}$$

Given the near-perfect ion selectivity of this electricity generator, it is plausible that enhancing the ion flux within the system could significantly improve its power density. Additional experiments were conducted to validate this hypothesis through two different approaches (these new data are present in Fig. S19 and Table S3).

1. Compact Device Configuration Design: Through the optimization of the device's layout or structure, our strategy achieved more efficient ion transport. This enhancement resulted in a significant increase in power density, boosting it by 3.4 times.

2. Increasing CO₂ Adsorption Capacity: Enhancing the CO₂ adsorption capacity through increased nanosheet loading significantly improved the device's ability to adsorb CO₂. This

enhancement led to a greater ion concentration gradient, thereby elevating the ion flux. As a result of these efforts, we achieved an approximate tenfold increase in the device's power density.

Please refer to the following data for the additional experimental works.

Supplementary Table S3. The power density of the devices calculated based on Fig. S17

Devices	Configuration	Nanosheet concentration	Volumetric peak power density ($\mu\text{W}/\text{cm}^3$)
a	6cm*2cm*0.5cm	1 mg/mL	0.0027
b	3cm*0.5cm*0.1cm	1 mg/mL	0.011
c	3cm*0.5cm*0.1cm	5 mg/mL	0.024

The revised manuscript now includes the new experimental data and corresponding discussion, as detailed below.

The practicality of NAH electricity generator was evaluated in terms of energy storage, conversion efficiency, scalability, response time, and modular integration. The generated electricity can be stored by charging a commercial capacitor (0.5 μF) (Fig. 4a). When connected to an external circuit with a 0.5 $\text{M}\Omega$ resistor during an adsorption cycle, the generator reaches its peak total energy generation of 24 $\mu\text{W}\cdot\text{h}/\text{mol}$ with a volumetric density of 0.0027 $\mu\text{W}/\text{cm}^3$ (Fig. 4b), corresponding to an energy conversion efficiency of approximately 0.6% relative to the total input energy (Supplementary Notes S6, 7 and Supplementary Figs. S15, S16). Although the energy conversion efficiency aligns with those of recently reported moisture electricity generators, the present peak density falls short of the required threshold^{20,24}. Given that near-perfect selectivity has been achieved, the bottleneck issue affecting the current low power density appears to be the insufficient ion flux in the system. Therefore, to improve the power density of NAH electricity generator, we pursued two distinct strategies: optimizing the device configuration for improved ion transport efficiency and enhancing CO_2 adsorption capacity for a higher concentration gradient. These two approaches achieved an overall 10-fold improvement in the power density. Future research is encouraged to focus on boosting the ion flux while maintaining the high selectivity to further increase efficiency and thus consolidate the practical implications (Supplementary Figs. S17, Table S3.). (Main text Page 11)

Reviewer #2 (Remarks to the Author):

General comments: Carbon dioxide capture and utilization is one of the key issues toward the sustainable development of society. This manuscript focused on carbon dioxide adsorption-based electricity generation, and presented a quite unique strategy for selective ion transport, by spatially nanoconfined separation of HCO_3^- anions and h-BN nanosheet- NH_3^+ cations with huge size differences, to achieve efficient power generation. Different from ion-exchange membranes, neutral nanoporous hydrogel matrix was used to spatially hinder h-BN nanosheet- NH_3^+ movement while allow HCO_3^- transport freely. In general, the manuscript was well

organized and the idea was fairly verified by experiments and calculations. The following major points need to be properly addressed ahead of publication.

Authors: We are grateful for the reviewer's constructive feedback and positive remarks on our manuscript. In response, we have made revisions and conducted additional work, as detailed below, to address and incorporate the insightful suggestions.

Specific comments: **Q1.** Is the sandwiched configuration necessary for the NAH electricity generator? I understand that the pure agarose hydrogel section is used for ion selection and separation. However, the role of the NAH hydrogel at the ion receiving reservoir side seems unclear. What happens if it is replaced by water?

Authors: The NAH hydrogel in the ion-receiving part of our device was originally intended to create a symmetrical structure between the two electrical collectors. This design choice was made to ensure that any electrical signals observed could be confidently attributed to the exposure of one part (the ion-releasing part) to gaseous CO₂. Following the suggestion, we conducted an additional experiment where we replaced the NAH hydrogel at the receiving part with pure DI water, while keeping other experimental conditions constant. This change to an asymmetric structure resulted in an initial open-circuit voltage of approximately 35 mV. We hypothesize that this voltage may arise from CO₂ adsorption by the nanosheet solution from ambient air during the device's fabrication process (Fig. R1).

Fig. R1 Power generation from CO₂ by the device with DI water as the ion receiving part.

We have included a relevant description in the caption of Figure 1 in the revised manuscript to provide further clarity on this aspect.

Fig. 1. Rational design of the NAH electricity generator from CO₂ adsorption. a, Schematic illustration of the prototype NAH electricity generator. The generator was intentionally designed with a symmetric structure between two electrical collectors to ensure a balanced chemical potential across the generator at the initial stage. (Man text, page 4)

Q2. What is the influence of the thickness of the pure agarose hydrogel on the ion transport rate and the power density? Usually, shorter diffusion path of ions facilitates higher ion flux and power density. It is recommended that the authors need to clarify this by experimental verifications.

Authors: Thank you for this valuable suggestion. In our new experiments, we controlled the length of the diffusion path and examined both the I_{SC} and V_{OC} under the same power generation process. Referring to the equation $qD_1A \frac{dC_1}{dx} - qD_2A \frac{dC_2}{dx}$ in supplementary note S5, the I_{SC} roughly equals the net ion flux difference of negative and positive ions. Since the diffusion rate of the positive ions is notably lower than that of negative ions, the I_{SC} can predominantly reflect the diffusion flux of negative ions. Hence, by monitoring the I_{SC} and V_{OC} , we can effectively analyze the impact of the diffusion path length on both the ion diffusion flux and power density.

We found shortening the diffusion path led to an increase in the peak ion flux as reflected by the I_{SC} curve, while it also accelerated the charging and discharging process of the generator. Nonetheless, the diffusion path shows negligible influence on the peak power density.

A relevant discussion has been incorporated into has been added to the revised manuscript.

Further experimental verification was conducted by controlling the length of the ion diffusion path, specifically by adjusting the width of the pure hydrogel in the middle (Supplementary Fig. S12). Reducing the width from 3 cm to 1 cm resulted in an increase in the peak I_{SC} , escalating from 140 nA to 169.7 nA, indicating a higher peak ion diffusion flux. Moreover, it achieved the peak I_{SC} in only 11.4 minutes compared to the 20.4 minutes required by the prototype of 3 cm. Additionally, the time taken to decay to half its peak I_{SC} decreased to 24.6 minutes from 40.2 minutes. This expedited charging and discharging process, achieved through

the reduction of the diffusion path, agrees well with the characteristics of ion-diffusion-induced power generation. (Main text, page 11)

Supplementary Fig. S14. Power generation in terms of I_{sc} (right) and V_{oc} (inset) by devices featuring different lengths of the ion diffusion path. The results demonstrate that a device with a shorter diffusion length exhibits a faster charging and discharging process compared to one with a longer diffusion path. The short circuit current (I_{sc}) suggests a higher peak ion flux achieved by shortening the diffusion path. However, despite these variations, their peak power density ($P=I_{max}*V_{max}$) remains at similar level. This observation implies that power density is more likely determined by the CO_2 adsorption capacity of the NAH hydrogel on the ion-releasing part rather than the pure hydrogel in the middle. (Supplementary page 22)

Q3. What is the maximum or optimal concentration of h-BN-NH₂ nanosheets that can be used for a best power generation performance?

Authors: Theoretically, the calculation suggests that the capacity for CO_2 adsorption correlates with the concentration gradient (Supplementary Nost S6 and S7). According to this theoretical framework, a higher concentration of h-BN-NH₂ nanosheets should result in an increased adsorption capacity, leading to improved performance. However, practical experiments have revealed only marginal performance improvement when increasing the concentration from the current 5 mg/mL to 10 mg/mL (Supplementary Fig. S10). One plausible explanation for this

minimal improvement could be the self-aggregation or restacking of nanosheets at higher concentrations. This phenomenon might hinder the adsorption capacity, counteracting the expected performance enhancement. Moreover, the surface area of the hydrogel device may act as a limiting factor, particularly at higher nanosheet concentrations. This limitation might affect both the kinetic and overall capacity of CO₂ adsorption.

The revised manuscript now includes new data and related discussion as outlined below.

Supplementary Fig. S10. The impact of h-BN-NH₂ concentration in the NAH hydrogel on power generation performance. The results demonstrated a substantial enhancement in power generation performance with an increase in concentration from 1 mg/mL to 5 mg/mL, yielding an approximate 61% improvement in peak V_{OC}. This aligns with the ionization formula provided in Supplementary Note S5, suggesting that the quantity of released ions is directly proportional to the amount of h-BN-NH₂ nanosheets. However, upon further increasing the concentration to 10 mg/mL, the generated power showed only marginal improvement, amounting to less than 5%. To enhance the performance, breakthroughs in material modifications are encouraged. Strategies such as increasing the PEI grafting ratio of the nanosheets or reducing the size of the nanosheets by using 2D quantum dots could potentially improve CO₂ adsorption capacity and water dispersion solubility, thereby enhancing overall performance. (Supplementary Page 18)

Increasing the concentration of h-BN-NH₂ nanosheets in NAH hydrogel from 1 mg/mL to 5 mg/mL resulted in an increase of V_{OC} from 90.4 mV to 145.7 mV with an extended half-value

period of V_{oc} from 3.5 h to 5.4 h (Fig. 2c). No further significant enhancement was achieved by increasing the concentration to 10 mg/mL (Supplementary Fig. S10). (Main text page 7)

Q4. There are some language mistakes in the manuscript that need to be corrected thoroughly. For example, in Figure 3f, the unit of vertical axis.

Authors: Thank you for pointing this out. The typos have been corrected.

Q5. During the pH swing process, can the CaCO_3 precipitant be removed completely from the hydrogel matrix? Otherwise, will it block the nanopores of the matrix and hinder the performance of the generator?

Authors: According to the Bjerrum plot, the CO_3^{2-} can transition into HCO_3^- within the pH range of 6-9, and transform into CO_2 (aq) + H_2CO_3 when the pH is below 4 (Fig. S4). Consequently, the CaCO_3 precipitate can theoretically be eliminated by immersing the generator in an acidic buffer solution.

In our experiments, no obvious precipitation was observed during the regeneration process, which we believe is due to the relatively low concentration of nanosheets in the hydrogel (at the milligrams per milliliter level). Besides, the hydrogel network is extremely porous with over 95% of its space serving as water-filling ion channels. Consequently, the minimal quantity of CaCO_3 precipitate did not have any noticeable adverse impact on the overall regeneration performance, as evidenced by the consistent peak V_{oc} , I_{sc} , and power density across 5 cycles (Fig. 2b and Supplementary Fig. S4b).

To make it clearer, the following revisions have been made in the revised manuscript.

The observed I_{sc} decay necessitates a proper regeneration method to ensure the reusability of NAH electricity generator. Referring to the Bjerrum plot (Supplementary Fig. S9a), the adsorbed CO_2 presents mainly as HCO_3^- ions in the pH range from 6.0-9.0. Considering the aqueous environment of the NAH electricity generator, the commonly adopted pH-swing strategy should be able to remove CO_2 and regenerate NAH electricity generator^{9,10}. During the pH swing desorption process, HCO_3^- ions can transition into CO_3^{2-} ions if the pH value is increased above 10.0, allowing sequestration by reacting with Ca^{2+} to form CaCO_3 precipitates¹¹. Alternatively, lowering the pH value below 4.0 converts HCO_3^- ions into CO_2

gas, which can be harvested for utilization¹². The regeneration was conducted by immersing the generator in an alkaline buffer solution (pH=10). The outcomes demonstrate that the I_{sc} and V_{oc} signals can be fully restored to their initial levels via pH swing after five adsorption-and-discharging cycles (refer to the last regeneration cycle in Fig. 2b and Supplementary Fig. S9b) (Main text, page 6-7)

Supplementary Fig. S9. a, Bjerrum plot, indicating the carbon species in water solution versus pH conditions. [13], Copyright 2008. Adapted with permission from Springer Nature. **b**, The V_{oc} testing after 5-cycle of short circuit discharging and pH swing regeneration, respectively. A buffer solution with a pH value of 10 was employed for the regeneration process. It's crucial to note that the resulting CaCO_3 was retained within the generator without undergoing any specific treatment, as the electrical signals can be fully restored. (Supplementary page 9)

Q6. Generating electricity from the adsorption of CO_2 has also been reported in recent works (e.g., <https://doi.org/10.1016/j.greenca.2023.08.002>). For a comprehensive literature review, these works should be cited and their contribution to this field should be properly acknowledged.

Authors: Thank you for providing this reference. Regrettably, it was not published at the time of our initial manuscript submission. We have now cited this paper in our revised manuscript, with corresponding adjustments made to both the introduction and conclusion sections.

In commercially available CO_2 chemical adsorption, the amine solutions can generate oppositely charged ions upon CO_2 absorption as follows: $\text{CO}_2 + \text{R-NH}_2 + \text{H}_2\text{O} \rightleftharpoons \text{R-NH}_3^+ + \text{HCO}_3^-$ (Equation 1), if these ions are effectively separated, they also suitable for electricity

generation as demonstrated recently^{1,10,11}. However, the positively charged amine ions and negatively charged bicarbonate ions generated in the conventional CO₂ adsorption process possess similar sizes at the molecular level and are well mixed, making their efficient separation challenging¹⁴⁻¹⁶. Dependence solely on a positively charged separation membrane yielded a mobility selectivity of 2.5 between HCO₃⁻ and R-NH₃⁺ ions¹, a significantly lower magnitude compared to what was attained in the osmotic energy generation process¹⁷. Addressing this challenge could potentially facilitate the pursuit of efficient energy conversion and sufficient power density in this promising field. (Main text Page 2)

7. Although the NAH electricity generator works, its current and the output power density is quite low, which may hinder its practical applications. Is there any way to improve the energy conversion efficiency? I encourage the authors to provide a suitable perspective and discussion in the conclusion part.

Authors: Reviewer 1 posed a similar query. In response, we carried out additional experiments focused on enhancing the power density of our device. These improvements were achieved through two distinct strategies: 1) optimizing the device configuration for compactness, and 2) increasing the CO₂ adsorption capacity. These combined efforts successfully resulted in a tenfold increase in the overall volumetric power density. For more detailed information, please refer to our response to the **general comment part B from Reviewer 1**.

Reviewer #3 (Remarks to the Author):

General comments: This is an interesting study of electricity generation from CO₂ absorption in an agarose hydrogel. The key idea is that the mobilities of the anion and cations generated by CO₂ absorption can be differentiated by coating the CO₂-absorbing PEI polymer onto nanoplatelets like BN or graphene sheets. The cationic species generated by the CO₂ absorption is then bound to the relatively large nanoplatelet (e.g., BN or graphene sheets) while the smaller anion is free to diffuse through the hydrogel. This is a novel idea, to my best knowledge. The authors convincingly demonstrate that the nanogenerators do produce electrical power as a result of the difference in mobilities of the ions through the hydrogel. I find the paper to be

interesting, well written, and a potentially valuable contribution to the literature on energy conversion and CO₂ capture. I do have the following comments and suggestions:

Authors: We sincerely appreciate the reviewer's positive feedback on our manuscript and the constructive suggestions aimed at enhancing its quality. Below, we detail the revisions and additional work we have undertaken, all of which are in response to the valuable questions provided.

Specific comments:

Q1a. Some further analysis of the electricity generation would be instructive, as the open-circuit osmotic voltage across the hydrogel can be directly related to the ion selectivity. Specifically, the transference number t_+ or the ratio of cation-to-total-ion flux is related to the V_{osm} as:

$$t_+ = J_+ / (J_+ + J_-) = (1/2) \{ V_{osm} / [(RT/zF \ln ((\gamma_H C_H) / (\gamma_L C_L)))] + 1 \}$$

where $\gamma_{H,L}$, R , T , z , F , are the activity coefficient, ion concentration, gas constant, temperature, valence, and Faraday constant, respectively. The subscript H and L would refer to the high- and low- ionic concentration sides across the hydrogel. For this particular study, one could assume that the mobility of the cation is negligible, so that $t_+ \rightarrow 0$. It seems then that the open circuit osmotic voltage would depend approximately as $\ln(C_H/C_L)$. Could the authors use the data shown in Fig. 2c&d to test this?

Authors: We appreciate the reviewer for this great suggestion. Given the challenge of quantifying the actual ion concentration gradient across the generator, it is difficult to calculate the transference number, selectivity, and osmotic potential without enough assumptions. To simplify it, we conducted additional experiments based on Fig. 2c&d. Referring to the recommended methodology, we directly tested the precursor h-BN-NH₂ solutions prepared for Fig. 2c within an H-cell setup. Using a hydrogel bridge, we separated a high-concentration nanosheet solution from a low-concentration solution. Employing a linear scanning voltage ranging from -100 mV to 100 mV across the H-cell with a pair of commercial Ag/AgCl electrodes, we measured the reversal potential, enabling us to calculate the transference number of cations and anions, as well as selectivity ratio (anion/cation). Our results highlighted the direct correlation between Open Circuit Voltage and the concentration gradient, they also

revealed a near-perfect charge selectivity, consistent with our earlier observations from diffusion tests.

Additional description and discussion were added in the revised manuscript:

We also monitored the I - V curves of h-BN-NH₂ water solutions after CO₂ adsorption in a H-cell, with a 3-fold concentration gradient separated by pure hydrogel. From this, a transference number of anion (t_-) nearing 1 was derived. This outcome further indicates the near-perfect anion/cation selectivity achieved by the hydrogel channels (Supplementary Fig. S12, Note S5).
(Main text, page 10)

Supplementary Note S5. Reversal potential and transference number

The h-BN-NH₂ (10h milling) water solutions of varying concentrations underwent a 1-hour CO₂ bubbling for testing purposes. These solutions were contained within an H-cell and were separated by a 2% hydrogel bridge, measuring 0.8 cm in diameter and 3 cm in length. In this setup, one compartment received 15 mL of a high concentration infusion (3 mg/mL and 6 mg/mL, repetitively), while the other compartment was supplied with 15 mL of a low concentration solution (1 mg/mL and 2 mg/mL, creating a threefold concentration gradient). To measure the I - V responses across the hydrogel bridge, an electrochemical station was connected to both compartments via salt-bridged Ag/AgCl electrodes at a temperature of 296.15 K. The reversal potential (E_R) was determined at the zero current point on the I - V curve. Using the provided potential equation below, we can calculate the membrane transference number (t), a pivotal indicator of ion selectivity⁷:

$$t_- = \frac{1}{2} \left[\frac{E_R}{\frac{RT}{zF} \ln \left(\frac{a_H C_H}{a_L C_{HL}} \right)} + 1 \right]$$

Where R represents the gas constant, T is the absolute temperature, z denotes the electrovalence of the counter ion (with z being 1 for positive ions and -1 for negative ions), C stands for the ion concentration within the solution, F represents the Faraday constant, H and L indicate high concentration and low concentration respectively, and t signifies the transference number. A transference number of $t_- = 1$ indicates an ideal anion-selective transport scenario, where no cation is permitted to pass through. **(Supplementary, Page 4)**

Q1b. By increasing the concentration of h-BN-NH₂ or the PEI grafting ratio, they would essentially be increasing CH. In my back-of-the-envelope estimation, going from an h-BN-NH₂ concentration of 1 to 5 mg/ml would lead to an expected $\ln(5)=1.6x$ increase in open-circuit voltage. This is pretty close to what is seen from the measured data in Fig. 2c. However, the increase in PEI grafting ratio shown in Fig. 2d seems to give a greater-than-expected increase in V_{osm} . Could the authors comment on why increasing the PEI grafting ratio seems to increase the electricity generation more than both theory and increasing h-BN-NH₂ concentration?

Authors: To investigate the unexpected surge in V_{osm} , as illustrated in Fig. 2d, we conducted additional experiments using h-BN-NH₂ nanosheet solutions with various PEI grafting ratios (Fig. R2). By assuming a transference number of anions (t^-) of 1, we calculated the ion concentration gradient (a_H/a_L), which is summarized in Table R1. Intriguingly, we found that the ion concentration gradient does not simply align with the PEI grafting ratio gradient. In fact, it consistently surpasses the latter, and this deviation becomes more pronounced as the grafting ratio increases. This led us to theorize that there might be additional amino groups, aside from

the grafted PEI, contributing to CO₂ absorption and ion release within the hydrogel, thereby expanding the actual ion concentration gradient.

The literature in materials science indicates that ball milling introduces edge functionalities¹⁸. Specifically, in the case of h-BN, the intense grinding action from milling balls fractures the B-N bonding within nanosheets, creating freshly exposed edges adorned with active B and N atoms. Subsequent contact with water triggers a hydrolysis reaction, transforming these atoms into -BOH and -NH₂ moieties¹⁹.

In our experiments, augmenting the PEI grafting ratio was achieved by prolonging milling times. Extended milling duration induces repeated in-plane breakages, generating smaller nanosheets with increased edges and, consequently, enhanced edge functionalities. This phenomenon has been explored in our previously published work earlier this year²⁰. Hence, we surmise that the **introduction of additional edge -NH₂ functionalities through prolonged milling** may positively contribute to the actual ion concentration gradient, surpassing those estimated according to the PEI grafting ratio gradient.

Fig R2. I-V curve recorded under different PEI grafting ratio gradient

Table R1. Comparison of the estimated ion concentration gradient and the PEI grafting ratio gradient

Testing solution (1mg/mL)	PEI grafting ratio gradient	Estimated ion concentration gradient

1h h-BN-NH ₂ and 5h h-BN-NH ₂	4.1	4.84
1h h-BN-NH ₂ and 15h h-BN-NH ₂	6.13	7.61

Q2. The I-V data in Fig. 3c is puzzling. Why do the I-V curves for the h-BN-NH₂ solution not go through the origin BEFORE CO₂ absorption? In fact, the increase in V_{oc} and I_{sc} after CO₂ absorption is relatively small compared to what exists beforehand. That would seem to imply that the hydrogel was ion selective and generating significant electrical power even before CO₂ absorption. Could the authors comment on this?

Authors: We believe that this misleading information could be mainly contributed by unintended CO₂ adsorption from the air during the device fabrication and testing process. To precisely assess the impact of CO₂ adsorption on the solution's electrical resistance and osmotic voltage, we did some optimization on this test by bubbling the nanosheet solution with N₂ and comparing it with CO₂. While the line still does not perfectly pass through the origin (V_{oc}<10 mV), this approach substantially reduces the influence of adsorbing CO₂ from the air, offering a clearer comparison. Revisions have been made in the revised manuscript and experiment description in Supplementary Note S3.

Fig. 3c, I-V curves of h-BN-NH₂ nanosheet water solution bubbled with CO₂ and N₂ respectively in a H-cell. The other side was fed with DI water.

The release of ions from adsorption is further substantiated by a threefold higher ion conductivity observed in the nanosheet solution bubbled with CO₂ compared to N₂. (Fig. 3c and Supplementary Note S3). **(Main text Page 8)**

Q3. Were the redox potentials at the electrodes subtracted from the measured potentials to get the true V_{osm} ?

Authors: The V_{osm} values presented in Fig. 2c and 2d were derived by calculating the difference between the values before and after CO_2 adsorption while operating under an open circuit condition. It's important to note that the initial value may not inherently be zero but often falls within a range typically less than 5 mV. Additionally, comparable values were obtained across multiple electrode materials such as Ag, Pt, and carbon. Consequently, we believe that the calculated V_{osm} values should be contributed by CO_2 adsorption.

Q4. It would be helpful to estimate the areal or volumetric power density of the generator. While I suspect that it would be low compared to other technologies like conventional reverse-electrodialysis membranes, the power density would still be a useful benchmark for future studies of CO_2 absorbing nanogenerators. On a related note, I do not understand the units on the “energy density” shown in Fig. 4b. Is it really “microW hr mol”?

Authors: Volumetric density has been provided in the revised manuscript as requested. We apologize for the earlier error where we incorrectly referred to the total generated power as the power density. In our previous calculation, $24 \mu W \cdot h/mol$ means generator can produce $24 \mu W \cdot h$ of energy by absorbing 1 mol CO_2 . The primary purpose of calculating it (energy normalized by the CO_2 adsorption) is to estimate the energy conversion efficiency. To enhance clarity and ease of understanding, we have made the following revisions to our manuscript:

When connected to an external circuit with a $0.5 M\Omega$ resistor during an adsorption cycle, the generator reaches its peak total energy generation of $24 \mu W \cdot h/mol$ with a volumetric density of $0.0027 \mu W/cm^3$ (Fig. 4b), corresponding to an energy conversion efficiency of approximately 0.6% relative to the total input energy (Supplementary Notes S6, 7 and Supplementary Figs. S15, S16). (Main text, page 11)

The generated electricity (W) can be calculated according to the current curve as a function of time monitored by the source meter:

$$W = \int RI^2(t) dt \quad (8)$$

Where R is the external resistance.

The total generated energy normalized by the theoretical CO₂ adsorption capacity (E_M) is calculated as equation 9.

$$E_M = \frac{W}{C_{Adsorption}} \quad (9)$$

Where $C_{Adsorption}$ is the maximum CO₂ adsorption capacity of the nanosheets within the hydrogel.

The volumetric power density (P_V) is obtained by equation 10.

$$P_V = \frac{I_M^2 R}{V} \quad (10)$$

Where I_M stands for the peak current. V is the total volume of the hydrogel.

The generated electricity (E_M) reached its peak at 23.9 $\mu\text{W}\cdot\text{h}/\text{mol}$ when connected to a 0.5 M Ω resistor, equivalent to 0.086 J/mol. The total energy input from the CO₂ adsorption calculated based on equation 7 is at 14.3 J/mol. This indicates an estimated energy conversion efficiency of approximately 0.6% at 293.15 K, comparable to the efficiency of bilayer polyelectrolyte films used in moisture power generation ($\sim 1\%$)²¹. The volumetric power density according to equation 10 stands at 0.0027 $\mu\text{W}/\text{cm}^3$. To enhance this relatively insufficient volumetric power density, increasing the device's CO₂ adsorption capacity and adopting a more compact design are recommended as future research directions. (Supplementary Page 7)

Q5. Why are there “glitches” as V_{oc} rises initially in Fig.s 4a, e, f? This may also appear in the data shown in Fig. 2a, although it is harder to see.

Authors: We have conducted a detailed analysis of the anomalies, referred to as "glitches," observed during our experiments. These were identified as fluctuations that consistently appeared during the ascending phases of V_{oc} . Notably, these fluctuations manifested at varying points across different external circuits. For instance, they were observed around the working potential of 1.6 V in the light-emitting diode (as illustrated in Fig. R3a), and approximately at 50 mV during the charging phase of a commercial capacitor (0.5 μF , depicted in Fig. R3b).

From these observations, we deduce that the charging speed may experience variations around the working potential of various external circuits. This variability is likely the cause of the "glitches" appearing at distinct points on the voltage ascending curve.

Fig. R3 The V_{oc} curve of charging the light-emitting diode (a) and the capacitor (b).

To clarify and avoid any future misunderstandings, we have implemented suitable revisions in the manuscript.

Fig. 4f, Generators in parallel and series (5×10) were used to power a light-emitting diode, the blue and red background indicate the status of the switch (on and off). The fluctuations on the Voltage ascending curve likely indicate that the charging speed is influenced by the characteristics of the external circuit. (Main text, Page 12)

Some minor points:

Q1. In Fig. 2b, how long was the regeneration cycle? Can it be pointed out in the data? Is it perhaps the small bump before $t=50\text{hr}$? And was it done with the generator in situ, or was the hydrogel removed, soaked, and then replaced? Some clarification here would be appreciated.

Authors: The generator was disconnected and subjected to a regeneration process and re-connected again for one more testing cycle. The small pump in Fig. 2d was caused by this connection and reconnection process. We added a small subsection detailing the regeneration process in the method part of the revised manuscript.

Generator regeneration. The regeneration of the generator was explored by choosing a pH-swing strategy to study its multiple utilization potential¹¹. Following 5 cycles of electricity

generation, the generator was removed and immersed in a calcium hydroxide (Ca(OH)₂) solution with a pH value of 10 for 1 hour. This was followed by washing the generator in excessive DI water 3 times. Subsequently, the generator was placed back in the testing box and subjected to the same testing procedures. (Main text page 16)

Q2. Voc and Isc are mis-labeled in Fig 3c.

Authors: They have been corrected.

Q3. What is “lambda_D” of HCO₃⁻ on page 10?

Authors: The term λ_D specifically denotes the hydrated ionic radius. Clarifications have been incorporated where necessary.

Q4a. The diffusion experiments discussed on pg. 10 are a little unclear. Was the hydrostatic difference that built up due to water transport considered or can it be argued to be negligible? In Fig. 3f, the image on the right shows that the column of water on the h-BN-NH₂ side increased in height over the course of 96 hours.

Authors: Thank you for bringing up this interesting question. To clarify the calculation process, an additional equation has been included in Supplementary Note S4 as follows:

The diffusion rate D is calculated based on the following equation.

$$D = \frac{C_t V_t}{A}$$

Where C_t is the measured concentration of sodium bicarbonate or h-BN-NH₂ nanosheet at time t , V_t is the volume of the solution in the DI water side at time t . A is the cross-section area of the hydrogel bridge. (Supplementary page 4)

Regarding the hydrostatic difference, it can be argued that it is negligible when compared to osmotic pressure. For instance, considering a 5mm water level difference, the hydrostatic pressure is estimated to be approximately 49 Pa. However, the osmotic pressure exerted by the NaHCO₃ solution used in our diffusion experiments is significantly higher. Based on the osmotic pressure formula (R1), the osmotic pressure ($\Delta\pi_{theo}$) value for this solution is around

2.93 kPa, which illustrates the substantial difference between hydrostatic and osmotic pressures in this context.

$$\Delta\pi_{theo} = M \cdot R \cdot T \quad (R1)$$

Where M is the total molar mass of the draw solution (including cation and anion) (mol), R is the gas laws constant (0.0821 L·atm/(mol·K)) and T is the testing kelvin temperature (298.15 K).

Q4b. Also, how was the concentration measured for the sodium bicarbonate experiments? Was the diffusion of Na^+ accounted for? When the authors say that the diffusion of sodium bicarbonate was $1.8 \cdot 10^6$ times faster than that of h-BN-NH₂, are they using the transport rate of HCO_3^- only, or of both Na^+ and HCO_3^- ? It would seem fairer to compare only the rate of HCO_3^- to that of h-BN-NH₂.

Authors: The concentration of sodium bicarbonate (in mol/mL) was determined using a conductivity meter to measure the solution's conductivity. A standard curve correlating solution conductivity with concentration was established beforehand. The transportation rate of HCO_3^- (in mol/m²) was approximated to be equivalent to the transportation rate of sodium bicarbonate (in mol/m²). This estimation was based on two reasonable assumptions: (1) the hydrogel channels exhibited no selectivity for transporting sodium ions or bicarbonate ions, and (2) the calculation didn't consider the influence of different counter ions (Na^+ and h-BN-NH₃⁺ in this case).

Corresponding revisions have been made:

The sodium bicarbonate diffusion rate is determined by assessing the conductivity variation within the DI water side concerning diffusion time using a conductivity meter (DZS-708TP, Shanghai Precision Scientific Instrument Co., Ltd). Considering that the agarose channel is significantly larger than both Na^+ and HCO_3^- ions, it is plausible to suggest that the channel lacks selective transport capacity for Na^+ or HCO_3^- ions. Consequently, the transport rate of HCO_3^- ions is inferred to be closely aligned with the transport rate of sodium bicarbonate. Please note that this estimation does not account for the influence of counter ions. In real cases, the difference in diffusion between h-BN-NH₃⁺ and HCO_3^- can generate a reverse electrical field, slowing down the following diffusion of the HCO_3^- ions. (Supplementary page 3)

Q5. The units of the interaction energies of the various ions with the agarose chains seem to be off. Are they supposed to be KJ/mol rather than KJs as currently given in the manuscript?

Authors: Thank you for pointing this out. These units have been corrected into KJ/mol.

Reference

- 1 Chen, C. *et al.* Nanofluidic osmotic power generation from CO₂ with cellulose membranes. *Green Carbon* **1**, 58-64 (2023).
- 2 Qian, Y. *et al.* Boosting osmotic energy conversion of graphene oxide membranes via self-exfoliation behavior in nano-confinement spaces. *J. Am. Chem. Soc.* **144**, 13764-13772 (2022).
- 3 Zhang, Z. *et al.* Cation-selective two-dimensional polyimine membranes for high-performance osmotic energy conversion. *Nat. Commun.* **13**, 3935 (2022).
- 4 Chen, C. *et al.* Bio-inspired nanocomposite membranes for osmotic energy harvesting. *Joule* **4**, 247-261 (2020).
- 5 Xin, W. *et al.* Biomimetic nacre-like silk-crosslinked membranes for osmotic energy harvesting. *ACS nano* **14**, 9701-9710 (2020).
- 6 Liu, Y.-C., Yeh, L.-H., Zheng, M.-J. & Wu, K. C.-W. Highly selective and high-performance osmotic power generators in subnanochannel membranes enabled by metal-organic frameworks. *Science Advances* **7**, eabe9924 (2021).
- 7 Tunuguntla, R. H. *et al.* Enhanced water permeability and tunable ion selectivity in subnanometer carbon nanotube porins. *Science* **357**, 792-796 (2017).
- 8 Wang, L., Wang, Z., Patel, S. K., Lin, S. & Elimelech, M. Nanopore-based power generation from salinity gradient: why it is not viable. *ACS nano* **15**, 4093-4107 (2021).
- 9 Bui, M. *et al.* Carbon capture and storage (CCS): the way forward. *Energy Environ. Sci.* **11**, 1062-1176 (2018).
- 10 Kim, E. J. *et al.* Cooperative carbon capture and steam regeneration with tetraamine-appended metal-organic frameworks. *Science* **369**, 392-396 (2020).
- 11 Kang, J. M. *et al.* Energy-efficient chemical regeneration of AMP using calcium hydroxide for operating carbon dioxide capture process. *Chemical Engineering Journal* **335**, 338-344 (2018).

- 12 Jin, S., Wu, M., Gordon, R. G., Aziz, M. J. & Kwabi, D. G. pH swing cycle for CO₂ capture electrochemically driven through proton-coupled electron transfer. *Energy Environ. Sci.* **13**, 3706-3722 (2020).
- 13 Fontela, M., Velo, A., Brown, P. & Pérez, F. in *Marine Analytical Chemistry* 1-37 (Springer, 2022).
- 14 Prajapati, A. *et al.* Migration-assisted, moisture gradient process for ultrafast, continuous CO₂ capture from dilute sources at ambient conditions. *Energy Environ. Sci.* (2022).
- 15 Goepfert, A. *et al.* Carbon Dioxide Capture from the Air Using a Polyamine Based Regenerable Solid Adsorbent. *J. Am. Chem. Soc.* **133**, 20164-20167 (2011).
<https://doi.org:10.1021/ja2100005>
- 16 Lv, B., Guo, B., Zhou, Z. & Jing, G. Mechanisms of CO₂ capture into monoethanolamine solution with different CO₂ loading during the absorption/desorption processes. *Environ. Sci. Technol.* **49**, 10728-10735 (2015).
- 17 Kim, S. *et al.* Neuromorphic van der Waals crystals for substantial energy generation. *Nat. Commun.* **12**, 1-10 (2021).
- 18 Jeon, I.-Y. *et al.* Edge-carboxylated graphene nanosheets via ball milling. *Proceedings of the National Academy of Sciences* **109**, 5588-5593 (2012).
- 19 Lei, W. *et al.* Boron nitride colloidal solutions, ultralight aerogels and freestanding membranes through one-step exfoliation and functionalization. *Nat. Commun.* **6**, 1-8 (2015).
- 20 Wang, Z. *et al.* Scalable high yield exfoliation for monolayer nanosheets. *Nat. Commun.* **14**, 236 (2023).
- 21 Wang, H. *et al.* Bilayer of polyelectrolyte films for spontaneous power generation in air up to an integrated 1,000 V output. *Nat. Nanotechnol.* **16**, 811-819 (2021).

REVIEWERS' COMMENTS

Reviewer #1 (Remarks to the Author):

At this stage I have no further suggestions for this manuscript.

Reviewer #2 (Remarks to the Author):

The requested revisions have been adequately addressed by the authors, and the authors have revised the manuscript carefully. This manuscript could be accepted for publication in Nat. Commun.

Reviewer #3 (Remarks to the Author):

I appreciate the authors' responses to my questions. In particular, while the authors explain that it is difficult to quantify the actual ion-concentration gradient across the generator, they did perform additional experiments measuring the transference number with the precursor h-BN-NH₂ solution in a 3-fold concentration gradient separated by pure hydrogel. I also appreciated the comparison between the PEI grafting ratio gradient and the ion-concentration gradient estimated from the open-circuit voltage. The comparison is quite reasonable, and thus, while the authors' hypothesis that "introduction of additional edge -NH₂ functionalities through prolonged milling may positively contribute to the actual ion concentration gradient" may be correct, it may not be strictly necessary to explain the data. The authors' explanation that Voc is generated because of unintended CO₂ adsorption from the air during the device fabrication and testing process is supported by the greatly reduced voltage when N₂ is bubbled through the solution. And I appreciate the authors' calculation of the volumetric power density and clarification that the generator was removed for the regeneration cycle. Overall, I find the concept explored in the paper to be interesting.

Response Letter

Electricity Generation from Carbon Dioxide Adsorption Enabled by Spatially Nanoconfined Ion Separation

(NCOMMS-23-48799A)

Reviewer #1 (Remarks to the Author):

At this stage I have no further suggestions for this manuscript.

Authors: We are grateful again for the reviewer's previous constructive feedback on our manuscript.

Reviewer #2 (Remarks to the Author):

The requested revisions have been adequately addressed by the authors, and the authors have revised the manuscript carefully. This manuscript could be accepted for publication in Nat. Commun.

Authors: We thank you for your recommendation for publication.

Reviewer #3 (Remarks to the Author):

I appreciate the authors' responses to my questions. In particular, while the authors explain that it is difficult to quantify the actual ion-concentration gradient across the generator, they did perform additional experiments measuring the transference number with the precursor h-BN-NH₂ solution in a 3-fold concentration gradient separated by pure hydrogel. I also appreciated the comparison between the PEI grafting ratio gradient and the ion-concentration gradient estimated from the open-circuit voltage. The comparison is quite reasonable, and thus, while the authors' hypothesis that "introduction of additional edge -NH₂ functionalities through prolonged milling may positively contribute to the actual ion concentration gradient" may be correct, it may not be strictly necessary to explain the data. The authors' explanation that Voc is generated because of unintended CO₂ adsorption from the air during the device fabrication and testing process is supported by the greatly reduced voltage when N₂ is bubbled through

the solution. And I appreciate the authors' calculation of the volumetric power density and clarification that the generator was removed for the regeneration cycle. Overall, I find the concept explored in the paper to be interesting.

Authors: We thank you for your positive comments and recommendation for publication.